# Perovskite Crystals: Unique Pseudo-Jahn–Teller Origin of Ferroelectricity, Multiferroicity, Permittivity, Flexoelectricity, and Polar Nanoregions

**Isaac B. Bersuker** [1],* and **Victor Polinger** [2]

1   Department of Chemistry, University of Texas at Austin, Austin, TX 78712-1229, USA
2   Department of Chemistry, University of Washington, Seattle, WA 98195-1700, USA; vpolinger@msn.com
*   Correspondence: bersuker@cm.utexas.edu

**Abstract:** In a semi-review paper, we show that the local pseudo-Jahn–Teller effect (PJTE) in transition metal B ion center of $ABO_3$ perovskite crystals, notably $BaTiO_3$, is the basis of all their main properties. The vibronic coupling between the ground and excited electronic states of the local $BO_6$ center results in dipolar distortions, leading to an eight-well adiabatic potential energy surface with local tunneling or over-the-barrier transitions between them. The intercenter interaction between these dipolar dynamic units results in the formation of the temperature-dependent three ferroelectric and one paraelectric phases with order–disorder phase transitions. The local PJTE dipolar distortion is subject to the presence of sufficiently close in energy local electronic states with opposite parity but the same spin multiplicity, thus limiting the electronic structure and spin of the $B(d^n)$ ions that can trigger ferroelectricity. This allowed us to formulate the necessary conditions for the transition metal perovskites to possess both ferroelectric and magnetic (multiferroic) properties simultaneously. It clarifies the role of spin in the spontaneous polarization. We also show that the interaction between the independently rotating dipoles in the paraelectric phase may lead to a self-assembly process resulting in polar nanoregions and relaxor properties. Exploring interactions of PJTE ferroelectrics with external perturbations, we revealed a completely novel property—*orientational polarization in solids*—a phenomenon first noticed by P. Debye in 1912 as a possibility, which was never found till now. The hindered rotation of the local dipole moments and their ordering along an external field is qualitatively similar to the behavior of polar molecules in liquids, thus adding a new dimension to the properties of solids—notably, the perovskite ferroelectrics. We estimated the contribution of the orientational polarization to the permittivity and flexoelectricity of perovskite crystals in different limiting conditions.

**Keywords:** perovskite crystals; Pseudo-Jahn-Teller effect; ferroelectricity; multiferroicity; permittivity; flexoelectricity; polar nanoregions; orientational polarization

## 1. Introduction

There are innumerable publications devoted to the study of crystals with perovskite $ABO_3$ structure, reflecting the rich variety of their properties with wide-ranging applications in physics, chemistry, and materials science. In this paper, we show that there is a fundamental structural feature of such crystals related to the high, cubic-octahedral symmetry of their metallic B center, which defines their main properties. This feature is the pseudo-Jahn–Teller effect (PJTE). Its role in triggering the ferroelectricity of perovskite crystals was revealed first more than half a century ago [1] but gained a more developed form in recent decades [2–7]. Similar to the Jahn–Teller effect (JTE), the PJTE, under certain conditions, makes a high-symmetry configuration unstable with respect to

lower symmetry distortions, but distinguished from the JTE, it does not require electronic degeneracy. Instead, the two or more electronic states that mix under the nuclear displacements (vibronic coupling) may have any energy gap, provided that the other relevant parameters of the system are appropriate to obey the PJTE condition of "pseudodegeneracy" [8–11]. Most importantly, the JTE in degenerate states and the PJTE in nondegenerate states are the only source of spontaneous symmetry breaking (SSB) in polyatomic systems [8–10], while degeneracy and pseudodegeneracy are the only source of SSB in the whole spectrum of transformations from elementary particles to nucleus, to atoms, molecules, crystals, and phase transitions [12]. Emerging from the first principles, the JTE and PJTE compensate for the violation of the adiabatic approximation in defining polyatomic space configurations by the high-symmetry positions of the atoms.

The metallic B center in perovskites is in ideal conditions to exhibit a local PJTE. Indeed, as illustrated in the next section, the crystal is centrosymmetric, and for some of the electronic configurations of the B ion, it has a nondegenerate ground state with a relatively low-lying excited state of opposite parity (but the same spin multiplicity), which mix under vibronic coupling to produce the PJTE dipolar distortions (note that the JTE does not produce dipolar distortions in centrosymmetric crystals). This B center dipolar instability plays a key role in all the main properties of the appropriate perovskites (in some perovskites, the A center may play a similar role, not fully explored so far).

Below we illustrate these statements using the most studied perovskite, $BaTiO_3$, as a basic example. In Section 2, we demonstrate how the PJTE operates in the B center of the perovskite crystal, with numerical estimates of the parameters for this crystal. Section 3 shows how the local polar distortions induced by the PJTE cooperate in the crystal to produce its paraelectric and ferroelectric phases. We discuss also the relation between the predicted by the vibronic theory order–disorder nature of the phase transitions in ferroelectric perovskites and the (apparently) observed partial displacive components in some experimental observations. The presence of sufficiently close in energy local ground and excited states that produce the dipolar distortion strongly depends on the electronic structure of the transition metal B ion in the $BO_6$ cluster. It takes place only for a limited number of cases with unpaired electrons, thus limiting the necessary conditions of multiferroicity and revealing explicitly the role of spin in spontaneous polarization (Section 4). Section 5 is devoted to a novel, unique property of the perovskites with the PJTE, namely the orientational polarization. The almost free or hindered rotation of the local dipole moments results in strong orientational effects in interaction with external perturbations (similar to polar liquids), which enhances permittivity, flexoelectricity, electrostriction, etc., by orders of magnitude. The possible existence of orientational polarizable solids was indicated more than a century ago, but not found till the present work. Section 6 demonstrates how the PJTE-induced local dipolar distortions explain the formation of polar nanoregions and relaxor properties of ferroelectrics in a Gibbs free energy controlled self-assembly process.

In entirety, the origin of a bundle of properties (practically, all main properties) of perovskite ferroelectrics, notably $BaTiO_3$, is thus explained based on one fundamental feature, the local PJTE, and new properties are predicted and confirmed experimentally. This result, as a pattern and together with a variety of other similar generalizations [8–14], strongly supports the role of the PJTE as a unique tool in exploring molecular and solid-state properties. It is remarkable that the same JTE and PJTE ideas served also as a basis in the understanding of the formation of local polarons, bipolarons, and bipolaron-strips in perovskite cuprates that led to the discovery and explanation of the origin of high-temperature superconductivity [15–26]. Note also that the treatment of the ferroelectric and superconductivity phenomena based on the local vibronic coupling significantly deviates from the traditional band theory of crystals, because the disordered local (JTE and PJTE) distortions violate the translational symmetry.

## 2. The Pseudo-Jahn–Teller Effect in ABO₃ Perovskite Crystals: B-Center Instability, Mueller's ESR Experiments

As outlined in the Introduction, the high symmetry of perovskite $ABO_3$ crystals makes them outstanding with respect to a variety of important properties. Among them, properties related to electronic degeneracy and pseudodegeneracy, which are directly controlled by symmetry, seem to be most important. The cases of electronic degeneracy leading to the Jahn–Teller effect (JTE) in the local states and their cooperative interactions in crystals are relatively easily recognized and well-studied [9,10]. Less attention has been paid until recently to the pseudo-JTE (PJTE), which is not seen directly from the initial structural and electronic data of the crystal. Meanwhile, as shown in a series of works [1–7], the main properties of some important perovskite $ABO_3$ crystals are controlled by the PJTE.

Consider the local PJTE in the paraelectric phase of $BaTiO_3$. Very briefly, in cluster language for the octahedral unit $[TiO_6]^{8-}$ (Figure 1), the local PJTE emerges from the vibronic mixing of the ground state $A_{1g}$, formed by the fully occupied six HOMO ((highest occupied (HO) molecular orbitals (MO)) $t_{1u}$ and $t_{2u}$ (mostly oxygen $2p$ orbitals) and three LUMO (lowest unoccupied, LU) $t_{2g}$, mostly titanium $3d$ orbitals (a total of nine MOs with the one-electron energy levels shown in Figure 2), by the polar $t_{1u}$ type normal coordinates $Q_x$, $Q_y$, and $Q_z$. This PJTE $(A_{1g} + T_{1u}) \otimes t_{1u}$ problem [9,10] results in a $9 \times 9$ secular equation. Its solution yields the following adiabatic potential energy surface (APES) of the $TiO_6{}^{8-}$ cluster, obtained already in the first paper on the subject [1]:

$$U(Q) = \frac{1}{2} K_0 Q^2 - 2\left[ \sqrt{\Delta^2 + 2F^2\left(Q^2 - Q_x^2\right)} + \sqrt{\Delta^2 + 2F^2\left(Q^2 - Q_y^2\right)} + \sqrt{\Delta^2 + 2F^2\left(Q^2 - Q_z^2\right)} \right] \quad (1)$$

where $Q^2 = Q_x{}^2 + Q_y{}^2 + Q_z{}^2$, $2\Delta$ is the energy gap between the mixing electronic states, $K_0$ is the primary force constant for the $Q$ displacements (stiffness of the crystal without the vibronic coupling), and $F$ is the vibronic coupling constant ($H$ is the Hamiltonian),

$$F = \langle 2p_z(O) \left| \left( \frac{\partial H}{\partial Q_x} \right)_0 \right| 3d_{xz}(Ti) \rangle \quad (2)$$

In a recent, more rigorous treatment by the Green's functions approach [5], the local PJTE in the cluster unit was appended with the bulk crystal properties by taking into account the interaction of the Ti ion with the whole crystal via its electronic and vibrational bands. It improved the results by yielding appropriately band-averaged parameters instead of the local cluster ones.

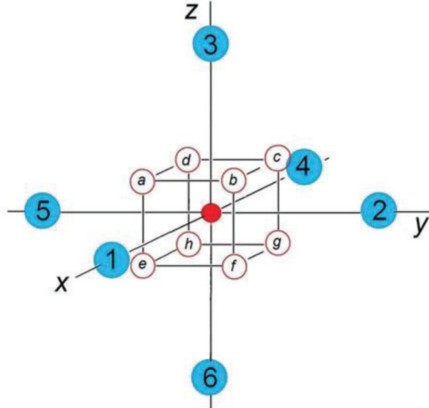

**Figure 1.** The octahedral fragment of the perovskite crystal structure $ABO_3$ with the transition metal atom B at the center (red) and six oxygen atoms at the apexes of the octahedron (numbered, blue). The letters *a*, *b*, *c*, etc., denote the induced by the PJTE eight equivalent off-center positions of the atom B in the eight wells of the APES (reprinted from [4]).

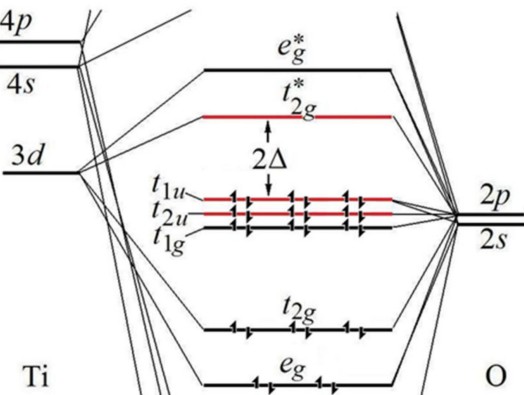

**Figure 2.** The energy-level correlation diagram (not to scale) for the octahedral cluster [TiO$_6$]. HOMO $t_{1u}$ and $t_{2u}$ (mostly oxygen) and LUMO $t_{2g}$ (mostly titanium) are one-electron energy levels of the orbitals that are mixed in the PJTE. The shown electron population corresponds to the particular case of [TiO$_6$]$^{8-}$ [4,10].

The three-dimensional APES (1) has a specific form (Figure 3). Under the condition

$$\Delta < 8F^2/K_0 \tag{3}$$

the surface (1) has a maximum (meaning instability) when the Ti ion is in the center of the octahedron, eight equivalent minima placed along the four trigonal axes, in which the Ti ion is displaced toward three oxygen ions (away from the other three); higher-in-energy 12 equivalent saddle points along the six $C_{2v}$ axes, at which the Ti ion is displaced toward two oxygen ions (at the top of the lowest barrier between two near-neighbor minima); and next six higher-in-energy equivalent saddle points, at which the Ti ion is displaced to one of the oxygen ions along the fourfold axes [1,2,5].

With an additional two experimentally determined structural constants, the band-averaged energy gap $2\Delta = 2.8$ eV, and the vibrational frequency at the bottom of the trigonal minimum $\hbar\omega_E = 193$ cm$^{-1}$, all the main parameters of this APES, shown in Table 1, were estimated [5], including K$_0$, F, the positions of the minima $Q_x = Q_y = Q_z = Q_0$ and first saddle points $Q_x = Q_y = q_0$, $Q_z = 0$, their PJTE stabilization energies, and the tunneling splitting $\delta_0$. The latter is a characteristic measure of the energy barrier between the near-neighbor minima of the APES. Its order of magnitude was first estimated by K. A. Muller and coworkers [27–29] in ESR experiments with probing ions by substituting Ti$^{4+}$ with the Mn$^{4+}$ ion, which produces a very similar APES (Mn$^{4+}$ is a "ferroelectric" ion, see below, Section 5), yielding approximate lifetimes in the minima (in sec.): $10^{-9} > \tau > 10^{-10}$. It is also seen in the NMR [30,31] and EXAFS [32,33] experiments: in the former, the characteristic "time unit" $\tau' \sim 10^{-15}$ and the Ti ion are seen in the trigonal minima in all the phases of BaTiO$_3$, while in NMR experiments, $\tau' \sim 10^{-8}$, and only an averaged picture is revealed (see the discussion in Section 3).

**Table 1.** Numerical values of the PJTE vibronic coupling and APES parameters of the Ti active centers in the BaTiO$_3$ crystal. $E_{PJT}$ [111] is the PJTE stabilization energy at the trigonal minima, $E_{PJT}$ [110] is the stabilization energy at the maximum of the barrier between two near-neighbor minima [5].

| $K_0$ (eV/Å$^2$) | $2\Delta$ (eV) | $F$ (eV/Å) | $\hbar\omega_E$ (cm$^{-1}$) | $Q_0$ (Å) | $q_0$ (Å) | $E_{PJT}$ [111] (cm$^{-1}$) | $E_{PJT}$ [110] (cm$^{-1}$) | $\delta_0$ (cm$^{-1}$) |
|---|---|---|---|---|---|---|---|---|
| 55 | 2.8 | 3.42 | 193 | 0.14 | 0.16 | −1250 | −1130 | 35 |

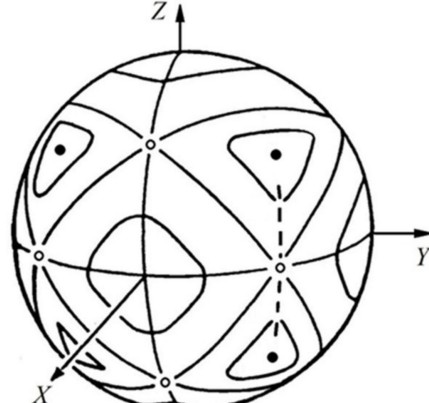

**Figure 3.** Contour map of the ground-state APES $U(Qx,Qy,Qz)$ for the octahedral cluster $[BO_6]^{8-}$ due to the PJTE coupling of its oxygen HOMOs to the central-atom's LUMOs. Equipotential level curves are projected on the surface of a sphere. Solid dots represent trigonal (rhombohedral) minima, open dots are orthorhombic saddle points. Broken curve is the path of steepest descent from one of the saddle points to the closest wells [6].

The four phases of $BaTiO_3$ emerge directly from this APES by gradually populating the states in the minima with temperature and stepwise overcoming its barriers [1,5,6] (see Section 3). Of particular interest is the prediction of disorder in the orthorhombic and tetragonal ferroelectric phases and in the cubic paraelectric phase, and order–disorder phase transitions between them, discussed below in more detail.

## 3. Ferroelectricity in ABO₃ Perovskite Crystals–Order–Disorder Phase Transitions

Due to the mentioned above high symmetry, perovskite crystals like $BaTiO_3$ have an inversion center and no local dipole moments. For such systems, the previous dominant theories of ferroelectricity assumed that the ferroelectric distortion occurs as a result of the compensation of the local repulsions (resisting dipolar displacements) by long-range dipole–dipole attractions. In such "displacive" theories, it is assumed that the local charge-separating displacements are induced by a self-consistent interaction with displacements in the bulk.

On the other hand, it has been proven that the JTE and PJTE are the only source of spontaneous distortions of high-symmetry configurations of polyatomic systems (at $T = 0$). As shown in the previous section, in perovskites with an inversion center, spontaneous local polar distortions are possible due to the PJTE. This triggered the idea that the spontaneous polarization in ferroelectric crystals is due to the temperature-dependent ordering of considered above PJTE-induced local dipole moments [1], presently developed in the vibronic PJTE theory of ferroelectricity (see the review [2] and references therein). According to this theory, the local dipole moments should be present in the crystal before its spontaneous polarization (meaning in its paraelectric phase), in contrast to the displacive theories, where the dipole moments occur as a result of the spontaneous polarization.

At the time of the first paper on the vibronic theory [1], there were no experiments to test the drastic differences between the two theories. Later on, the number of experimental observations that contradicted the displacive theories of ferroelectricity gradually increased (continuing), including X-ray scattering and diffraction data [34], Raman and optical reflective experiments [35–38], the mentioned above ESR with probing ions [27–29], EXAFS measurements [32,33], NMR experiments [30,31], neutron scattering [39], and later in a variety of experimental studies (see more details and references in [2]; the authors of [34] did not cite the first prediction of the order–disorder phase transitions in ferroelectric perovskites [1] but acknowledged it in a personal letter; see [40]). In these experiments, several "peculiar" (for displacive theories) features in the ferroelectric properties were revealed in some ABO₃ perovskites, notably $BaTiO_3$. Among them, we emphasize here the instant trigonal

displacement of the Ti ion in BaTiO$_3$ in all its four phases, ferroelectric and paraelectric [32,33], in strong disagreement with displacive theories, in which the metal off-center displacement occurs as a result of the phase transition to the polarized phase. This fundamental property of perovskite ferroelectrics is confirmed by a variety of experimental data, including the above-mentioned ones, and actually by any observations that implicate the Ti ion. Not only is this ion instantly displaced along one of the [111]-type directions in the paraelectric phase, where the averaged symmetry is cubic, but it is as well displaced in this trigonal direction in the tetragonal phase, where the crystal symmetry and the macroscopic polarization are tetragonal [39]. "The most striking example is the off-center displacement of the Ti atom in barium titanate observed in the paraelectric phase way above the Curie temperature of the tetragonal-to-cubic phase transition. As accurate measurements indicate [32,33], the Ti ion remains instantly displaced closely along [111] directions *throughout all the four* BaTiO$_3$ *phases*, and the magnitude of the off-center displacement decreases monotonically by only 13% when heating from 35 to 590 K, showing *no steps at the phase transitions.*" [32,33].

All these experimental observations and other empirical data (see [2]), as a pattern and in details, confirm the predicted earlier picture of the special, induced by the PJTE local dipolar distortions, which in cooperative interactions produce all observed ferroelectric properties. In a more recent work [6], using the numerical estimates of Table 1, the interaction between the PJTE-induced local dipole moments in BaTiO$_3$ was taken into account in a mean-field approximation, in which both the local off-center displacement and the mean-field of the environment are interdependent in a self-consistent way (see below). It yields the experimentally observed phase transitions in BaTiO$_3$ with reasonable values of Curie temperatures [6].

The picture of the B center of the ABO$_3$ crystal with the PJTE-induced eight-minimum APES in Equation (1) and local dipolar displacement in the minima, and their changes under the influence of the external mean-field **E** of the environment, is illustrated in Figure 4.

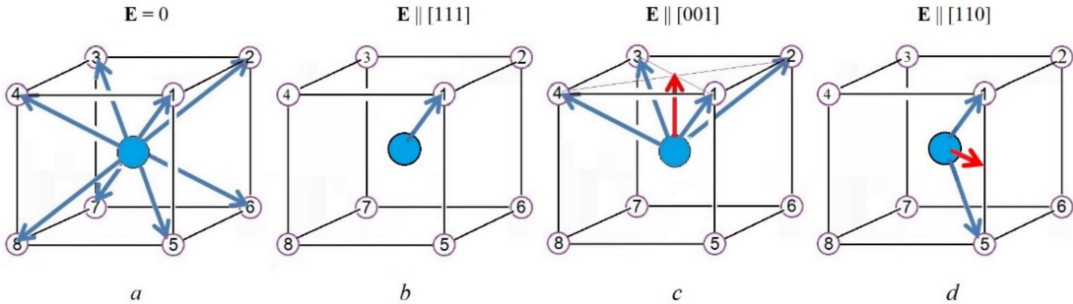

**Figure 4.** Polar displacements (blue arrows) of the central atom (blue) in the octahedron [BO$_6$]. White circles correspond to off-center equilibrium positions of the central ion B (the blue ball) in eight trigonal wells. (**a**) At no polar perturbation, the trigonal wells are symmetry-equivalent and therefore equally populated. The ion B has an equal probability to be off-center shifted (blue arrows) to each well and, on average, its off-center shift is zero. (**b**) Under a trigonal field **E** ∥ [111], one well is deeper than the others, and at low temperature, it is the only one temperature populated. The vector of average displacement (the blue arrow) is along the field parallel to the axis [111]. (**c**) In the tetragonal field **E** ∥ [001], the wells 1, 2, 3, and 4 remain symmetry-equivalent to one another; they are the lowest in energy and hence most populated. The mean displacement (red vector) points are along the axis [001]. (**d**) With the field E along the second-order axis, **E** ∥ [110], the wells 1 and 5 are the lowest and the only ones thermally populated. Therefore, the mean displacement (red vector) is along the electric field, **E** ∥ [110] [6].

Qualitatively, the phenomenon of ferroelectric ordering of the PJTE-induced local dipolar distortions is similar to magnetization in ferromagnetic crystals. The Hamiltonian of the system is $H = \sum_{\mathbf{m}} H_{\mathbf{m}} + \sum_{\mathbf{mn}} H_{\mathbf{mn}}$, where **m** and **n** label different unit cells of the lattice. Similar to Section 2, the one-cell term $H_{\mathbf{m}} = H_0(\mathbf{m})$ includes the PJTE in the octahedral unit [BO$_6$]. As for the inter-cell

coupling term $H_{\mathbf{mn}}$, we employ the simplified version of dipole–dipole coupling, which follows from the physical nature of ferroelectric ordering. In the mean-field **E** of all other dipoles in the crystal, the energy of a selected dipole $\mathbf{p_m}$ equals $-\mathbf{p_m} \cdot$E. At each site, the local dipole is influenced by the mean-field produced by the ordering of all other dipoles. At every other site, the induced electric dipole $\mathbf{p_n}$ is proportional to the applied electric field **E**. Therefore, $H_{\mathbf{mn}} = -\frac{1}{2} A_{\mathbf{mn}} \mathbf{p_m} \cdot \mathbf{p_n}$ with $\mathbf{m} \neq \mathbf{n}$. Due to the symmetry of the crystal lattice, the inter-cell coupling parameters $A_{\mathbf{mn}}$ are symmetric to swapping the indices, $A_{\mathbf{mn}} = A_{\mathbf{nm}}$. In what follows, the inter-cell constants $A_{\mathbf{mn}}$ are combined into one correlation constant, which is determined by fitting to experimental data.

To decouple $H_{\mathbf{mn}}$, we apply the mean-field approximation. Based on the smallness of the fluctuation $\Delta \mathbf{p_m} = \mathbf{p_m} - \langle \mathbf{p_m} \rangle$ from the average value $\langle \mathbf{p_m} \rangle$, it provides reasonable answers far from the phase transition when $|\Delta \mathbf{p_m}| << p_{\mathbf{m}}$ but does not apply close to the Curie temperature $T_C$. In the ferroelectric phase with a uniform polarization, the average $\langle \mathbf{p_m} \rangle$ is the same at different sites, so $\langle \mathbf{p_m} \rangle = \langle \mathbf{p} \rangle$. We insert $\mathbf{p_m} = \langle \mathbf{p} \rangle + \Delta \mathbf{p_m}$ into $H_{\mathbf{mn}}$ and neglect the terms quadratic in $\Delta \mathbf{p_m}$. Omitting the constant terms and plugging the resultant approximated expression into $H_{\mathbf{mn}}$ yields the decoupled Hamiltonian of the whole crystal as a sum of independent (mean-field) Hamiltonians, $H = \sum_{\mathbf{m}} H_{\mathrm{MF}}(\mathbf{m})$. In each of its ferroelectric phases, the perovskite crystal has translation symmetry. The one-site Hamiltonian is the same for different unit cells, $H_{\mathrm{MF}}(\mathbf{m}) = H_{\mathrm{MF}} = H_0 - \mathbf{p} \cdot$**E**. Here, $H_0$ is the one-site vibronic Hamiltonian yielding the PJTE in Section 2, and **E** is the mean-field, $\mathbf{E} = A \langle \mathbf{p} \rangle$, with the inter-cell correlation parameter $A = A(\mathbf{m}) = \sum_{\mathbf{n}} A_{\mathbf{mn}}$. Due to the translational symmetry, **E** is the same for all sites, so is $A = A(\mathbf{0}) = \sum_{\mathbf{m}} A_{\mathbf{m0}}$.

In the paraelectric phase with zero mean-field, $\mathbf{p} \cdot$**E** $= 0$, and there is no intercenter dipolar coupling. The one-site nuclear dipolar displacements are independent of the other sites. It reduces to uncorrelated hindered rotations of the polar distortions $\mathbf{Q_m}$ at different sites (via tunneling or over-the-barrier transitions between the equivalent minima of the APES), yielding the bulk average $\langle \mathbf{Q} \rangle = 0$, meaning a disorder of local dipole moments. Lowering the temperature below $T_C$ creates conditions for spontaneous polarization. Even a weak mean-field **E** can violate the equivalence between the wells and quench the tunneling; by locking the system in the lowest-in-energy potential wells and orienting the local dipoles in the direction **E** (Figure 4), it polarizes the crystal.

The effect of the mean-field depends on the magnitude of the PJTE induced dipole moment **p**. The latter is a combination of nuclear and electronic contributions, $\mathbf{p} = \mathbf{p}_{\mathrm{nucl}} + \mathbf{p}_{\mathrm{el}}$. It is easy to prove that due to the vibronic coupling, both $\mathbf{p}_{\mathrm{nucl}}$ and $\mathbf{p}_{\mathrm{el}}$ are proportional to the polar distortion **Q** induced by the applied electric field. Therefore, in what follows, we assume $\mathbf{p} = Z_B e \mathbf{Q}$, where $Z_B$ is the so-called Born charge, and $e$ is the elementary charge. The average value $\langle \mathbf{p} \rangle$ of the local dipole moment is proportional to the off-center nuclear displacement, $\langle \mathbf{p} \rangle = Z_B e \langle \mathbf{Q} \rangle$. Induced by the mean-field, it depends on $\langle \mathbf{Q} \rangle$ in a self-consistent way (see below). Therefore, we present the mean-field Hamiltonian as $H_{\mathrm{MF}}(\mathbf{m}) = H_{\mathrm{MF}} = H_0 - \lambda \mathbf{Q} \cdot \langle \mathbf{Q} \rangle$ with $\lambda = Z_B^2 e^2 A$.

In the lowest-temperature rhombohedral phase at $T = 0$ K, the strength of the mean-field is maximal, and the octahedral unit cell [$BO_6$] is locked in one of the trigonal wells at the bottom of the trough with the radius $Q_0$. Therefore, at $T = 0$ K, the magnitude of $\langle \mathbf{Q} \rangle$ approaches its maximum value $Q_0$. At a higher temperature $T \neq 0$ K, the nuclear motion delocalizes over different wells and the vector $\langle \mathbf{Q} \rangle$ decreases in magnitude and changes its direction. Therefore, for the sequence of the corresponding ferroelectric phase transitions, we take $\langle \mathbf{Q} \rangle / Q_0$ as the order parameter.

In a certain unit cell, the delocalization changes with temperature. Nuclear motion depends on the radius $Q_0$ of the trough and the height of potential barriers between trigonal wells. Both are due to the one-site PJTE (see Table 1 for $BaTiO_3$), whereas the mean-field constant $A$ is due to the inter-cell interaction, $A = \sum_{\mathbf{m}} A_{\mathbf{m0}}$. Generally, in cubic perovskites, there is a wide range of coupling parameters. In what follows, we consider two limiting cases: (*a*) deep trigonal wells where the tunneling model applies, and (*b*) shallow wells along the warped trough of the APES with the hindered rotation of the nuclei. The two cases correspond to significantly different mechanisms of delocalization and, correspondingly, of the ferroelectric phase transitions.

*a*   *The tunneling model for deep trigonal wells.* Most of the time, the nuclear motion is localized in the trigonal wells with low frequency tunneling through the barriers from one well into another. In a typical perovskite with PJTE, the tunneling energy gaps are of the order of several meV (see the estimates for $BaTiO_3$ in Table 1). The excited states of hindered rotations along the trough are a few dozen meV higher in energy. In such cases, the tunneling model applies, in which we can (approximately) consider just the eight tunneling states and neglect any quantum entanglement of all other excited states.

At $T > T_C$, in the paraelectric phase with zero mean-field, $E = 0$, self consistently, the order parameter is zero, $\langle \mathbf{Q} \rangle = 0$. The one-site Hamiltonian $H_{MFA} = H_0$ has cubic symmetry, and all trigonal wells are symmetry-equivalent. With no tunneling (assuming that the potential barriers are infinitely high), the ground state corresponds to eight Born–Oppenheimer states localized in the wells $|a\rangle$, $|b\rangle$, $|c\rangle$, ..., $|h\rangle$, labeled in Figure 1. Tunneling results in an even spread of the nuclear motion over the potential trough, lifting the eight-fold degeneracy of the ground state. The ground term splits into two triplets and two singlets, $A_{1g} + T_{1u} + A_{2u} + T_{2g}$ [41]. The tunneling splitting depends on the three overlap integrals, $\langle a|b\rangle$, $\langle a|c\rangle$, and $\langle a|g\rangle$. Evidently, for neighboring wells, $\langle a|b\rangle$ dominates. Due to the greater distance, in a good approximation, the other two, $\langle a|c\rangle$ and $\langle a|g\rangle$, can be neglected. Measured from the ground-state energy level $E_0$ in the infinite-deep wells, the tunneling energy levels are equidistant, $E(A_{1g}) \approx -3\Gamma$, $E(T_{1u}) \approx -\Gamma$, $E(T_{2g}) \approx \Gamma$, and $E(A_{2u}) \approx 3\Gamma$, where $\Gamma$ is the tunneling parameter,

$$\Gamma \approx \frac{\hbar \omega_E}{2\pi} \exp\left( -\frac{1}{\hbar} \int_v^w \sqrt{2m[U(s) - E_0]} ds \right) \tag{4}$$

$\hbar \omega_E$ being the energy quantum of the transversal mode in a trigonal well, the one perpendicular to the corresponding trigonal axis, $U(s)$ is the potential energy (the APES) in Equation (1), and $v$ and $w$ are the classical turning points where the system hits the barrier wall. The integral is taken over the arc length $s$ (broken curve in Figure 3) along the path of steepest descent from the orthorhombic saddle point to the closest trigonal minimum points of the APES. For $BaTiO_3$, the numeric estimate is $\Gamma \approx 35$ cm$^{-1}$ = 4.3 meV [5].

Within the basis set of the eight tunneling states, the one-site Hamiltonian $H = H_0 - \lambda \mathbf{Q} \cdot \langle \mathbf{Q} \rangle$ of the mean-field approximation is represented by the following matrix [41],

$$\mathbf{H}(\gamma) = \Gamma \begin{array}{c} \begin{array}{cccccccc} |A_{2u}\rangle & |T_{2g}\xi\rangle & |T_{2g}\eta\rangle & |T_{2g}\zeta\rangle & |T_{1u}x\rangle & |T_{1u}y\rangle & |T_{1u}z\rangle & |A_{1g}\rangle \end{array} \\ \begin{pmatrix} 3 & -\gamma_x & -\gamma_y & -\gamma_z & 0 & 0 & 0 & 0 \\ -\gamma_x & 1 & 0 & 0 & 0 & -\gamma_z & -\gamma_y & 0 \\ -\gamma_y & 0 & 1 & 0 & -\gamma_z & 0 & -\gamma_x & 0 \\ -\gamma_z & 0 & 0 & 1 & -\gamma_y & -\gamma_x & 0 & 0 \\ 0 & 0 & -\gamma_z & -\gamma_y & -1 & 0 & 0 & -\gamma_x \\ 0 & -\gamma_z & 0 & -\gamma_x & 0 & -1 & 0 & -\gamma_y \\ 0 & -\gamma_y & -\gamma_x & 0 & 0 & 0 & -1 & -\gamma_z \\ 0 & 0 & 0 & 0 & -\gamma_x & -\gamma_y & -\gamma_z & -3 \end{pmatrix} \end{array} \tag{5}$$

Here, $\gamma_j = \lambda Q_0 \langle Q_j \rangle / (\Gamma \sqrt{3})$ and $j = x, y, z$ is the dimensionless mean-field order parameter in terms of the tunneling constant $\Gamma$. The three order parameters $\gamma_j$ are solutions of the following system of coupled transcendental equations, $Z_\gamma \langle Q_j \rangle_\gamma = \text{Tr}(e^{-\beta H} Q_j)$ and $Z_\gamma = \text{Tr}(e^{-\beta H})$ with $j = x, y$, and $z$. Here, $\beta = 1/kT$, and the index $\gamma$ means self-consistent dependence on the components $\gamma_j$ included in the mean-field Hamiltonian (5). In the lowest-temperature rhombohedral phase, all atoms B are coherently shifted into the same trigonal well, say, in the direction [111]. In this case, $\gamma_x = \gamma_y = \gamma_z = \gamma$, and there is just one transcendental equation to solve. The mean-field **E** deepens the respective trigonal well. If **E** is strong enough, with the ordering energy $\mathbf{p} \cdot \mathbf{E}$ comparable to $\Gamma$, it quenches the tunneling and locks the system in this well. Rising temperature weakens the mean-field, reducing its locking effect.

For different values of the correlation parameter $j = p_0A/\Gamma$, the calculated temperature dependence of the order parameter $\gamma$ is shown in Figure 5. Plugging the resultant value of $\gamma$ into the expression for the Helmholtz free energy, $\Phi = -\partial(\ln Z_\alpha)/\partial\beta$ with $\beta = 1/kT$, we find the free energy $\Phi_{111}$ of the rhombohedral phase. Figure 6 shows the temperature dependence of $\Phi$ for different ferroelectric phases where we assume $j = p_0A/\Gamma = 5$.

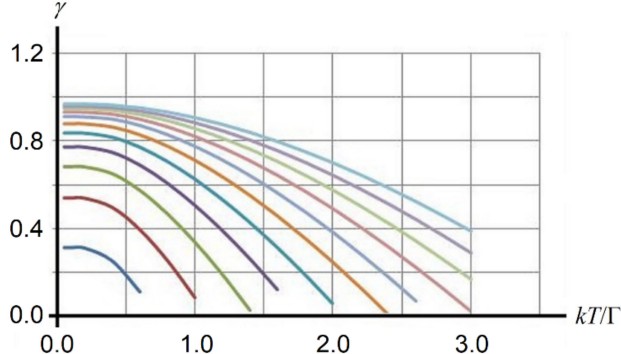

**Figure 5.** Temperature dependence of the order parameter $\gamma$ in the rhombohedral phase when the tunneling mechanism of disorder dominates. The correlation parameter, $j = p_0A/\Gamma$, varies from $j = 1$ for the left-bottom graph to $j = 3$ for the graph at the right top [6].

In the orthorhombic phase, all atoms B are off-center shifted along the two-fold symmetry axis [110]. As there is no minimum on the APES in this direction, we treat this "displacement" as resulting from an averaged motion evenly spread over two near-neighbor trigonal wells, the well $b$ and the well $f$ in Figure 1, separated by the orthorhombic potential barrier. In this case, $\gamma_x = \gamma_y = \gamma$ and $\gamma_z = 0$, and again we have just one equation to solve. Plugging the evaluated values of $\gamma_x = \gamma_y = \gamma$ and $\gamma_z = 0$ into the Helmholtz free energy, we find $\Phi_{110}$ in the orthorhombic phase (Figure 6). Solving the equation $\Phi_{111} = \Phi_{110}$ for temperature, we find $T_C$ for the rhombohedral-to-orthorhombic phase transition. At around $kT/\Gamma = 3.8$, the graph of $\Phi_{111}$ intersects with that of $\Phi_{110}$. Above $T_C = 3.8\Gamma/k$, the free energy $\Phi_{111}$ of the rhombohedral phase is above the orthorhombic phase $\Phi_{110}$. The latter becomes energy-advantageous, and therefore, at $T_C = 3.8\Gamma/k$, the rhombohedral-to-orthorhombic phase transition takes place. In this model, its dependence on the inter-cell coupling parameter $A$ is close to linear, $kT_C \approx 0.8p_0A - 0.3\Gamma$ [6].

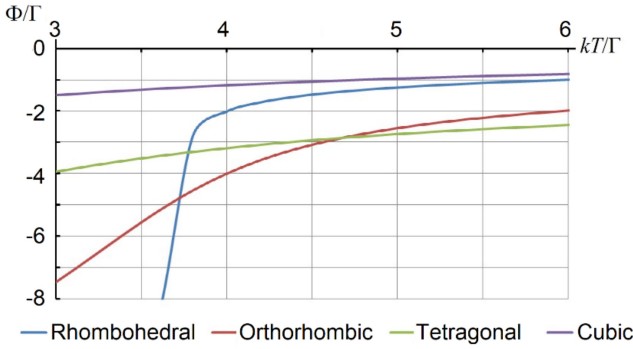

**Figure 6.** Temperature dependence of the Helmholtz free energy $\Phi$ (in units of $\Gamma$) for the four phases, rhombohedral, orthorhombic, tetragonal, and the paraelectric, at $j = 5$. At $kT < 3.8\Gamma$, the free energy of the rhombohedral phase is the lowest. With temperature, at $kT > 3.8\Gamma$, the free energy of the orthorhombic phase becomes the lowest, causing the rhombohedral-to-orthorhombic phase transition. Above $kT \approx 4.6\Gamma$, the tetragonal phase becomes most energy-advantageous [6].

In the tetragonal phase, all atoms B are off-center "shifted" to a tetragonal position, parallel to one of the four-fold symmetry axes, say [001]. Due to tunneling through four potential barriers, similar to the orthorhombic phase, this apparent shift is the average over four close-neighbor trigonal wells, *a*, *b*, *c*, and *d* in Figure 4, and the respective mean-field order parameter can be set as $\gamma_x = \gamma_y = 0$ and $\gamma_z = \gamma$. Plugging it into the expression of Helmholtz free energy $\Phi_{001}$ and solving the equation $\Phi_{110} = \Phi_{001}$, we find the temperature of the orthorhombic-to-tetragonal phase transition. For $j = 5$, it takes place at $kT_C \approx 4.7\Gamma$. Similar to the previous case, in this phase transition, the dependence of $T_C$ on the coupling parameter *A* is linear, $kT_C \approx p_0 A - 0.3\Gamma$ [6].

For BaTiO$_3$, the experimental values of the two low-temperature phase transitions, rhombohedral-to-orthorhombic $T_C$(I) and orthorhombic-to-tetragonal $T_C$(II), are $T_C$(I) = 178 K and $T_C$(II) = 278 K. Repeated with different values of the correlation parameter *A*, the tunneling model calculated values of $T_C$(I) = 204 K and $T_C$(II) = 256 K are closest to the experimental values when $j = 5$. The percentage errors are 15% and 8%, respectively. A lower value of the mean-field parameter *A* better describes the lower-temperature phase transition than the higher-temperature one. This result is quite understandable because of the larger amplitudes of vibration in the wells at higher temperatures and hence the stronger inter-cell interaction of the distortions.

Among the essential results of the tunneling model is the first-order type of ferroelectric phase transition. The equations which we used to evaluate Helmholtz free energy follow from the first-order perturbation approach to the eight-fold degenerate ground state. In this theory, the small parameter is the tunneling overlap $\Gamma$. In the case of barium titanate, as one can see from Figure 6, the tunneling model fails to provide a reasonable description of the high-temperature phase transition, tetragonal-to-cubic. It could happen by averaging over all eight trigonal wells. Such an averaging implies coherent tunneling through all twelve orthorhombic barriers. Obviously, due to the relatively high entropy factor, the respective probability is too low. An alternative tunneling path is through the high-symmetry point of the APES, which in the oxygen octahedron [BO$_6$] corresponds to the on-center position of the atom B. This potential barrier, of the order of 0.2 eV = 1600 cm$^{-1}$, is too high for tunneling.

*b.　Shallow trigonal wells, low potential barriers.* The warping of the two-dimensional trough on the APES may happen to be relatively weak, which results in moderately low potential barriers between trigonal wells, but the high-symmetry point, like in barium titanate, may be too high in energy, so the first-order tunneling does not hold. In this case, the high-temperature disorder is achieved by second-order processes through a temperature population of the excited states that are close in energy to the top of the corresponding potential barriers between near-neighbor wells. Such over-the-barrier hopping is similar to the Arrhenius-type activation in chemical reactions, and it resembles the Orbach relaxation in magnetic resonance.

The temperature of the ferroelectric phase transition $T_C$ is assumed high enough to populate the close in energy excited states, $kT_C \geq \Delta E_{JT}$. Then, the hindered rotation in the warped trough can be treated in terms of classical physics. The temperature averaging contains the exponent with the potential energy $U = U_0 - \lambda \mathbf{Q} \cdot \langle \mathbf{Q} \rangle$ of the mean-field Hamiltonian $H = H_0 - \lambda \mathbf{Q} \cdot \langle \mathbf{Q} \rangle$. Since it includes $\langle \mathbf{Q} \rangle$ as a parameter, we come to the following system of coupled transcendental equations:

$$Z\langle Q_j \rangle = \int Q_j e^{-U(\mathbf{Q})/kT} d^3 Q, \qquad Z = \int e^{-U(\mathbf{Q})/kT} d^3 Q \qquad (6)$$

In the tetragonal phase, the atom B is off-center shifted in one of the three tetragonal directions, say, along the axis [001]. In this case, $\langle Q_x \rangle = \langle Q_y \rangle = 0$ and $\langle Q_z \rangle = \langle Q \rangle \neq 0$. Out of the three equations (6), we have just one to solve. With the same parameter values as used above in the tunneling model, the numerical solution of this equation is shown in Figure 7. With temperature, the order parameter $\langle Q \rangle / Q_0$ decreases smoothly to zero at $kT_C \approx 0.1236 E_{JT}$. In the case of barium titanate with $F_0^2 / K_0 \approx 0.25$ eV, this corresponds to $kT_C \approx 250$ cm$^{-1}$ = 359 K, reasonably close to the experimental value of 373 K.

Thus, distinguished from the two low-temperature phase transitions, trigonal-to-rhombohedral and rhombohedral-to-tetragonal, the transition to the cubic phase is of second order.

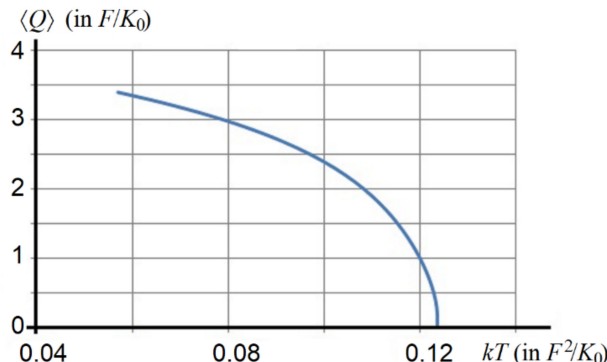

**Figure 7.** Temperature dependence of the order parameter $\langle Q \rangle$ (in units of $F/K_0$) in the tetragonal phase versus $kT$ (in units of $F_0^2/K_0$). At $kT \approx 0.123 \, F_0^2/K_0$, the off-center displacement of the atom B smoothly drops to zero, manifesting the second-order phase transition [6].

A fundamental conclusion, which was outlined already in the first paper of the vibronic PJTE theory [1], is that the ferroelectric phase transitions in such perovskite crystals are of order–disorder nature. This was first confirmed for $BaTiO_3$ in the experiments with diffuse X-ray scattering [32,33,40]. However, some more recent experimental findings—in particular, the low-frequency (soft) mode at the Curie temperature, observed in neutron scattering [42], IR absorption [43], hyper-Raman scattering [44], etc.—were interpreted as demonstrating displacive features of the phase transition (see also [45]).

As we noted earlier [2,6], there is no contradiction between these experimental observations and the vibronic PJTE theory. To begin with, the latter does not exclude the possibility of a low-frequency mode near the phase transition: it follows directly from the temperature dependence of the Helmholtz' free energy, $\Phi$, with respect to the limiting phonon TO mode, $[\omega_{\text{eff}}(T)]^2 = (\partial^2\Phi/\partial q^2)_0$. The consideration of the phase transition above in this section is based on the mean-field approximation which does not include any details about phonon dispersion. It follows from the EXAFS experiments [32,33] that in $BaTiO_3$, the magnitude of the local instant dipolar distortions does not change significantly as a result of the phase transition, meaning that the polarization process is similar to the magnetic ordering of local spins, which is of pure order–disorder type.

An attempt to include the phonon dispersion in a similar scheme involving the PJTE was undertaken in several papers [46–50], in which the PJTE is applied in a two-band approach, instead of our local two-state (in fact, several state) approach, outlined in Sections 2 and 3. The major difference between the two approaches is in the way the vibronic Hamiltonian is treated. In our paper, the mean-field approximation is used before any Fourier transformation to crystal waves is applied. It addresses the orientation degrees of freedom of the "pseudo spin" at each unit cell. Therefore, obviously, it provides a better description of the properties of local origin and the order–disorder phase transitions. In the two-band theory, the mean-field approximation is applied after the Jahn–Teller Hamiltonian is presented in crystal plane-wave form. It operates with crystal lattice modes that are uniform over the whole crystal, showing that the growing population of the low-energy portion of the electron energy band with lowering temperature strengthens the pseudo Jahn–Teller effect and reduces the respective curvature, $(\partial^2\Phi/\partial q^2)_0$. Evidently, this approach is adjusted to describe displacive transitions, but it fails to explain the variety of the experimental data of local origin [25–39]—notably, the instant trigonal displacements of the Ti ion in all the four phases, irrelevant to the phase transition [32,33].

The difference between the two approaches to the solid-state problem with the PJTE, local versus bulk (cooperative), is of a more fundamental nature than it may seem at first sight. Indeed, both the JTE and PJTE are of local origin: they are defined by off-diagonal matrix elements of the vibronic coupling terms in the Hamiltonian, which are significant only for the non-zero overlap of the electronic

wavefunctions of near-neighbor atoms. Therefore, if the interaction between the JTE or PJTE centers are not strong enough, meaning before their ordering (before the structural phase transition), their local JT dynamics are not correlated, the translation symmetry of the crystal is obeyed in average, but not in each moment of time, and the traditional presentation of the crystal structure by electron and phonon bands is inadequate. To our knowledge, there are no worked out general methods (or computer programs) to handle such systems beyond our approach, described above based on the cited papers.

In view of this significantly local origin of the PJTE, another important issue emerges with regard to the interpretation of the mentioned above experimental data [42–44] as showing elements of displacive phase transitions in $BaTiO_3$: it did not take into account the relativity to the means of observation (see [2,6,12]). Indeed, in the eight-minimum APES, the B ion under consideration has a characteristic "lifetime" $\tau \approx \hbar/\Gamma$, where $\Gamma$ is the introduced above tunneling parameter. In cubic perovskites, $\Gamma$ ranges from 0.01 to 50 $cm^{-1}$. Respectively, the characteristic time is from $\tau \approx 10^{-9}$ s = 1 ns to $\tau \approx 10^{-13}$ s = 0.1 ps. Similarly, since (before the ordering) the transitions between the minima at different centers are not correlated, there is a related characteristic dimension $l$, which is the size of one elementary cell. For barium titanate, it is of the order of $l \approx 4$ Å.

On the other hand, the experimental methods of observation have their own characteristic "time of measurement" $\tau_{exp}$, which is directly related to their frequencies. For example, the NMR technique measures the nuclear quadruple transitions and motional average dynamic displacements with a characteristic $\tau_{exp} = 10^{-8}$ s, whereas in EXAFS experiments, $\tau_{exp} = 10^{-15}$. All the other widely used experimental techniques lie in between these limits. Similarly, they have characteristic length limitations $l_{exp}$ mostly determined by the wavelengths. Except for the EXAFS measurements, $l_{exp}$ for all the other widely used experimental methods are several orders of magnitude larger than the unit cell dimension where the disorder begins. Obviously, the experimental results in [42–44] reflect the crystal processes averaged over many unit cells, both in space and time, and hence they cannot be used as an indication of displacive versus order–disorder phase transition.

It follows from this discussion that the most appropriate experimental method to reveal the microscopic origin of ferroelectric properties of perovskite crystals is EXAFS, and the already performed experiments with this method fully support the conclusions from the vibronic (PJTE) theory.

## 4. PJTE-Induced Orientational Polarization in Solids—Application to Dielectric Susceptibility and Flexoelectricity

One of the most important novel properties of ferroelectric $ABO_3$ perovskites, revealed by the vibronic (PJTE) theory, is their strongly enhanced interaction with external perturbations (electric fields, pressure, strain, etc.). As outlined in Section 3, in the absence of external influence, the PJTE-induced local dipolar distortions in $[BO_6]$ units are of dynamic nature, moving between the eight equivalent minima of its APES by tunneling or over-the-barrier transitions. In fact, the orthorhombic barriers between neighboring trigonal wells are relatively low, so the dynamics of the dipolar distortions may be regarded as some hindered rotations along a three-dimensional trough. In this respect, the crystal acquires properties of polar liquids, with all the consequences for the observable properties, including orientational polarization of crystals, first revealed in the vibronic PJTE theory [51–53]. Remarkably, the possible existence of dielectric crystals with randomly oriented dipole moments (similar to ferromagnetics), which may behave like a polar liquid above its freezing point, was first suggested by P. Debye [54] in 1912 (long before the experimental discovery of ferroelectricity), but not discovered till the present work. Similar to polar molecules in liquids, the orientational polarizability of solids with dynamic, PJTE-induced dipolar displacements is expected to be larger than displacive polarizability by orders of magnitude [51–53].

Different crystals vary with respect to the barrier height and the radius of the trough $Q_0$ (see Section 2). At high temperatures, $T > T_C$, in the paraelectric phase with zero mean-field, the trigonal wells are symmetry-equivalent, the tunneling evenly distributes the nuclear motion over the trough, and for the ion B, on average, its off-center shift $\langle \mathbf{Q} \rangle$ is zero (Figure 4a). The tunneling

splitting of localized states in eight trigonal wells is proportional to the tunneling parameter $\Gamma$. The necessary condition of tunneling is the resonance of the states, localized in the minima. In a typical ferroelectric perovskite with the PJTE, the energy gaps $2\Gamma$ between tunneling energy levels are of the order of magnitude of several meV. Higher in energy with a few dozen meV are excited states of hindered rotations along the trough (for $BaTiO_3$, the estimates are in Table 1). Any polar perturbation $W$, even not very strong, lowering one or several of the eight wells and/or raising the other ones violates the resonance, quenches the tunneling, and locks the system in the lowest well(s) (Figure 4b–d). The perturbation $W$ causes a non-zero off-center shift $\langle \mathbf{Q} \rangle$, which produces anisotropy in the dielectric and elastic properties. A remarkably small magnitude of $W \sim \Gamma$, of the order of, or even less than, meV, diminishes the tunneling and locks the system in the deepest minimum. A similar sensitivity, though to a lesser extent, is in the excited states of the hindered rotations. *This unique feature of the PJTE in cubic perovskites determines their significant response to polar perturbations resulting in orientational polarization* [51–53].

There are two types of polar perturbations: (*a*) electromagnetic fields targeting the electronic subsystem, and (*b*) applied strain distorting the crystal lattice. In what follows, we discuss both effects in ferroelectric perovskites. In both cases, we consider the paraelectric phase at $T > T_C$ with no mean-field and a relatively strong PJTE, when the tunneling approximation applies to the ground state. This allows us to reduce the problem to just eight tunneling states. For all the remaining excited states, we assume that their thermal population is small, and hence their contribution to the corresponding response function is negligible. The effect of locking the PJTE dipolar distortions is shown in Figure 8 [53], where we use an applied electric field $\mathbf{E}$ as an example of the polar perturbation, $W = -\mathbf{p} \cdot \mathbf{E}$, and the corresponding Hamiltonian is $H = H_0 - \mathbf{p} \cdot \mathbf{E}$, with $H_0$ as the Jahn–Teller Hamiltonian (Section 3). The magnitude of the induced dipole moment $\mathbf{p}$ is a combination of nuclear and electronic contributions, $\mathbf{p} = \mathbf{p}_n + \mathbf{p}_e$. As shown in [53], in the case of static applied field, the nuclear contribution dominates, $|\mathbf{p}_n| >> |\mathbf{p}|$. Therefore, as mentioned above, $\mathbf{p} \approx \mathbf{p}_n = Z_B e\mathbf{Q}$, where $Z_B$ is the so-called Born charge, and $e$ is the elementary charge constant. The induced off-center nuclear displacement $\langle Q \rangle$ is expected to approach its maximum value $Q_0$ under a trigonal field of infinite strength, when $E \rightarrow \infty$ and the octahedral unit cell $[BO_6]$ is entirely locked in one of the trigonal wells (Figure 4b). Accordingly, for the induced dipole moment, its asymptotic value is $p_0 \approx Z_B e Q_0$. When $\mathbf{E} \parallel [110]$, the perpendicular component averages out, $\langle Q_z \rangle = 0$, and the asymptotic value is $\langle Q \rangle \rightarrow (Q_0\sqrt{6})/3 \approx 0.8Q_0$. Similarly, under the tetragonal field $\mathbf{E} \parallel [001]$, the perpendicular components average out, $\langle Q_x \rangle = \langle Q_y \rangle = 0$, with the asymptote value $Q_0/\sqrt{3} \approx 0.6Q_0$. The general outcome for $\langle Q \rangle$ (in units of $Q_0$) as a function of $E$ (in units of $\Gamma/p_0$), shown in Figure 8, was obtained by diagonalizing the Hamiltonian matrix $8 \times 8$ with the subsequent thermal population of the resultant vibronic states [53].

## 4.1. Dielectric Susceptibility

In a cubic perovskite, in its paraelectric phase with zero mean-field and no applied field, $E = 0$, the crystal is not polarized. Under an applied electric field, a non-zero dipole moment $\mathbf{p}$ is induced in each unit cell, and the crystal becomes polarized. By definition, the vector of polarization $\mathbf{P}$ is the average dipole moment per unit volume, $\mathbf{P} = \langle \mathbf{p} \rangle/a^3$. Here, $a$ is the lattice constant, $\langle \mathbf{p} \rangle = \mathrm{Tr}(p_i e^{-\beta H})/\mathrm{Tr}(e^{-\beta H})$, with the averaging over both the temperature and (statistical) over different unit cells, $H = H_0 - \mathbf{p} \cdot \mathbf{E}$ is the vibronic Hamiltonian of the octahedral site $[BO_6]$ in the applied electric field, and $\beta = 1/kT$. Regularly, there is a linear relationship between the generalized "displacement" $\mathbf{P}$ and the "force" $\mathbf{E}$, namely $P_i = \varepsilon_0 \Sigma_j \alpha_{ij} E_j$. It corresponds to the first terms in the power expansion of $\mathbf{P}$ in terms of $\mathbf{E}$. The linear factor $\alpha_{ij}$ is the rank-two tensor of dielectric susceptibility or polarizability.

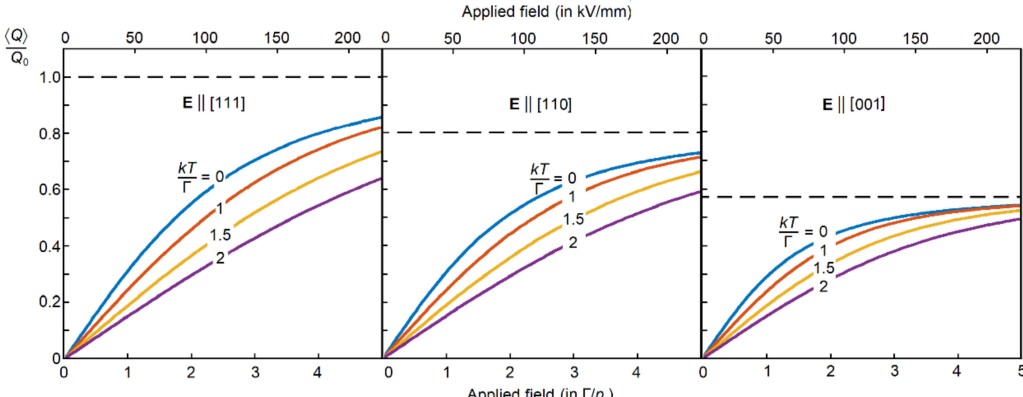

**Figure 8.** Approximate values of $\langle Q \rangle / Q_0$ versus the applied electric field $E$ (in $\Gamma/p_0$) resulting from the tunneling Hamiltonian at three different directions of the applied electric field, (**a**) **E** $\|$ [111], (**b**) **E** $\|$ [110], and (**b**) **E** $\|$ [001], and at four different temperatures, $kT = 0$, $\Gamma$, $1.5\Gamma$, and $2\Gamma$. The broken lines are horizontal asymptotes when $E \rightarrow \infty$. Shown above is the scale corresponding to the case of barium titanate with $\Gamma = 35$ cm$^{-1}$ [53].

In cubic crystals with no PJTE, the three principal values of the tensor $\alpha_{ij}$ are equivalent, and the polarizability is the same in all directions. The dielectric properties of such a crystal are isotropic, and the principal axes have no special direction. As follows from the numeric results in Figure 8, to be specified below, in cubic perovskites, the PJTE brings about two important properties: (1) the dielectric susceptibility has cubic anisotropy; the principal axes are "tied" to the symmetry axes [100], [010], and [001], and (2) the induced dielectric susceptibility is non-linearly dependent on the applied electric field. The polarizability, and, correspondingly, the dielectric permeability, is not a constant but depends on the strength of the applied field.

a.   *Low potential barriers and/or relatively weak electric fields, $E << \Gamma/p_0$.* This limiting case of $E \rightarrow 0$ corresponds to the left side of Figure 8, close to its vertical axis where the field-dependence on $\langle Q \rangle$, and hence on $P$, is linear. The consideration can be limited to just one localized state in each trigonal well (the potential trough is two-dimensional; even in a very shallow well, there is at least one localized quantum state). Under the applied field, the wells are not exactly symmetry-equivalent. However, since the electric field is relatively weak, the field-induced symmetry breaking is not significant. The localized states in trigonal wells are close to the resonance, with an even distribution of the nuclear motion over the trigonal wells, and the unit-cell octahedron [BO$_6$] carries frequent tunneling transitions or/and hindered rotations close to the bottom of the trough. Accordingly, assuming $P_i = \varepsilon_0 \Sigma_j \alpha_{ij} E_j$, we have $(\partial P_i / \partial E_j)_0 = \varepsilon_0 \alpha_{ij}$, where the subscript "0" means the derivative at $E = 0$. Plugging $\langle p_i \rangle = \text{Tr}(p_i e^{-\beta H}) / \text{Tr}(e^{-\beta H})$ into the definition $\mathbf{P} = \langle \mathbf{p} \rangle / a^3$, we come to $\alpha_{ij} = \alpha \delta_{ij}$ with

$$\alpha \approx \frac{p_0^2}{3\varepsilon_0 a^3 \Gamma} \frac{\sinh(3\Gamma\beta) + \sinh(\Gamma\beta)}{\cosh(3\Gamma\beta) + 3\cosh(\Gamma\beta)} = \frac{Z_B^2 e^2 Q_0^2}{3\varepsilon_0 a^3 \Gamma} \frac{\sinh(3\Gamma\beta) + \sinh(\Gamma\beta)}{\cosh(3\Gamma\beta) + 3\cosh(\Gamma\beta)} \tag{7}$$

Hence, in a weak electric field, the polarizability is isotropic and the principal axes of the tensor $\alpha_{ij}$ have no specific direction. At relatively low temperatures and/or large $\Gamma$, when $kT << \Gamma$ (meaning $\Gamma\beta \rightarrow \infty$), the temperature factor in Equation (7) approaches unity, and $\alpha \approx p_0^2/(3\varepsilon_0 a^3 \Gamma)$. At high temperatures and/or relatively narrow tunneling gap $\Gamma$, when $kT >> \Gamma$ (meaning $\Gamma\beta \rightarrow 0$), the temperature factor in Equation (7) approaches $\Gamma/(kT)$, and $\alpha \approx p_0^2/(3\varepsilon_0 a^3 kT)$. To be expected, this result coincides with the classic Langevin–Debye equation for orientational polarizability of dipolar molecules [55]. For BaTiO$_3$, assuming $Z_B \approx 7.8$ and $Q_0 \approx 0.19$ Å, we have $p_0 \approx 7$ D. Therefore, at $a \approx 4$ Å and $T \approx 500$ K, the Equation (7) gives $\alpha \approx 33$, close to the polarizability of some polar liquids.

b.  *Strong electric fields and/or deep potential wells*, $E \geq \Gamma/p_0$ (this case apparently does not apply to $BaTiO_3$, with $\Gamma \approx 35$ cm$^{-1}$ and $p_0 \approx 5$ D, $E = \Gamma/p_0 = 46$ kV/mm, which is above its dielectric breakdown). In this case, as follows from Figure 8, the field-dependence of $\langle Q \rangle$ and, hence, of $\langle p \rangle$ is non-linear. Therefore, for the dielectric susceptibility, instead of the derivative $(\partial P_i/\partial E_j)_0$, we can use the average $\alpha_{ij} = (\varepsilon_0)^{-1}(\Delta P_i/\Delta E_j)$. Since $\mathbf{P}(0) = 0$, we find $\Delta P_i/\Delta E_j = (P_i - 0)/(E_j - 0) = P_i/E_j$. For the applied electric field pointing along the trigonal axis, $\mathbf{E} \parallel [111]$, we have $E_x = E_x = E_x = E/\sqrt{3}$. According to Figure 4b, it lowers the trigonal well 1 in Figure 1, lifts the opposite well 7, and keeps the remaining potential wells unchanged. At low temperatures, the dipole moment $\langle Q_x \rangle = \langle Q_y \rangle = \langle Q_z \rangle \approx Q_0/\sqrt{3}$, so that $\langle Q \rangle = Q_0$. In this case, the polarization $P \approx \langle p \rangle/a^3 = p_0/a^3$. Similar results can be obtained for the cases when the applied electric field points along the symmetry axes [110] or [001]. The only difference is in the corresponding magnitude of $\langle p \rangle$. For $\mathbf{E} \parallel [110]$, we find $\langle p \rangle \approx 2p_0/\sqrt{6}$ whereas for a tetragonal field, $\mathbf{E} \parallel [001]$, we have $\langle p \rangle \approx p_0/\sqrt{3}$. Thus, for the case of a strong electric field, we come to

$$\alpha[111] \approx \frac{Z_B e Q_0}{\varepsilon_0 a^3 E}, \alpha[001] = \frac{\alpha[111]}{\sqrt{3}} \approx \frac{Z_B e Q_0}{\varepsilon_0 a^3 E \sqrt{3}}, \alpha[110] = \frac{2\alpha[111]}{\sqrt{6}} \approx \frac{2 Z_B e Q_0}{\varepsilon_0 a^3 E \sqrt{6}} \tag{8}$$

This angular dependence can be approximated by the cubic invariant,

$$\alpha(E, \varphi, \theta) \approx \frac{12 Z_B e Q_0}{\varepsilon_0 a^3 E} [0.8 - 0.5 Y_4(\varphi, \theta) + 2.8 Y_6(\varphi, \theta)] \tag{9}$$

where $Y_4(\phi, \theta)$ and $Y_6(\phi, \theta)$ are the spherical harmonics, $Y_4(\theta, \varphi) = (x^4 + y^4 + z^4 - \frac{3}{5}r^4)/r^4$ and $Y_6(\theta, \varphi) = x^2 y^2 z^2/r^6$. As follows from Figure 8, if the applied electric field is strong enough, the system is locked at the bottom of the lowest wells of the APES, and the polarization approaches its asymptotic value shown by the dotted line in Figure 8. Therefore, in all three formulas of Equation (8), the polarizability is inversely proportional to $E$, approaching zero at $E \to \infty$.

c.  *High temperature and/or deep potential wells*, $kT \gg \Gamma$. This case is of special interest. In most ferroelectric perovskites, the Curie temperature $T_C$ is of the order of a few dozen meV, while $\Gamma$ is only a few meV. Therefore, for most cubic perovskites in the paraelectric phase, the condition $kT \gg \Gamma$ applies. Tunneling energy gaps are of the order of $2\Gamma$. Therefore, in the high-temperature case when $\beta \to 0$, the Boltzmann exponent $\exp(-\beta H)$ can be approximated as $1 - \beta H$, and the temperature average $\langle \mathbf{p} \rangle = \mathrm{Tr}(\mathbf{p}e^{-\beta H})/\mathrm{Tr}(e^{-\beta H})$ simplifies $\langle \mathbf{p} \rangle \approx \mathrm{Tr}[\mathbf{p}(1 - \beta H)]/\mathrm{Tr}(1 - \beta H) \approx (-\beta/8)\mathrm{Tr}(\mathbf{p}H)$. At this point, one can take advantage of the invariance of the trace of a matrix to any unitary transformations of the basis set. In particular, we can use the basis of the eight tunneling states when $\mathbf{E} = 0$ with the well-known matrices $8 \times 8$ for the dipole moment $\mathbf{p}$ and the Hamiltonian $H$. Therefore, under any applied field, $\mathrm{Tr}(\mathbf{Q}) = 0$, and $\mathrm{Tr}(\mathbf{p}H) = -8p_0\mathbf{E}/3$. Hence, at relatively high temperatures in the applied field of any strength, we have $\langle \mathbf{p} \rangle \approx p_0^2/(3kT)$, resulting in the Langevin–Debye equation $\alpha \approx p_0^2/(3\varepsilon_0 a^3 kT)$. The dielectric susceptibility is isotropic with the arbitrary direction of the principal axes. The orientational contribution to the dielectric susceptibility $\alpha$ caused by the PJTE-induced dipole moment rotations in the trough in the limits of high temperatures and deep potential wells is thus inversely proportional to temperature. However, the experimentally measured value of $\alpha$ also includes the significant contribution of the PNR (see Equation (16) in Section 6).

### 4.2. Flexoelectricity

Similarly, in cubic perovskites, the PJTE produces an enhanced response to another kind of dipolar perturbation, the strain induced by applied forces, which acts directly upon the crystal lattice. Caused by the stress, the corresponding strain is a rank-two symmetric tensor $u$ with components $u_{mn} = \frac{1}{2}(\partial U_m/\partial x_n + \partial U_n/\partial x_m)$, where $\mathbf{U}$ is the vector of deformation, excluding the rigid-body motions. We follow the traditional notation $x_n$ with the index values $n = 1, 2,$ or 3 corresponding to rectangular

coordinates $x$, $y$, and $z$. In centrosymmetric crystals, the parity of $u$ is even. Therefore, in the paraelectric phase, a uniform strain does not cause any polarization. A non-uniform strain with a non-zero gradient $\partial u_{mn}/\partial x_j \neq 0$ is odd, and hence it can polarize the crystal. Like in the electric-field case, any non-uniform strain, even a weak one, breaks the resonance between the tunneling states, quenches the tunneling, and locks the system in lowest well(s) (Figure 4). The polarization **P** induced by the strain gradient is called the flexoelectric effect [56–58]. The stronger the perturbation $\partial u_{mn}/\partial x_j$, the greater the induced polarization. As shown in Figure 8, if strong enough, it can induce a non-linear polarization with a cubic anisotropy. In what follows, we limit the consideration with a relatively weak strain gradient $\partial u_{mn}/\partial x_j$ and its linear response.

Expanding polarization **P** in terms of $\partial u_{mn}/\partial x_j$, we can keep just the first non-zero terms,

$$P_i = \sum_{k,m,n} f_{ikmn} \frac{\partial u_{mn}}{\partial x_k} \tag{10}$$

Similar to the influence of an external electric field, due to the break of the local inversion symmetry induced by the PJTE, a relatively small strain gradient can induce a measurable polarization. The rate of change of $P_i$ with the strain gradient $\partial u_{mn}/\partial x_k$ may be called flexoelectric susceptibility. The strength of the flexoelectric coupling is determined by the four-rank tensor $f$ with components $f_{ijmn}$, called flexotensor. The elements $\partial u_{mn}/\partial x_j$ of the strain gradient can be treated as components of the rank-three tensor $u' = \nabla \otimes u$.

In the cubic symmetry group $O_h$, the components of the vector operator $\nabla$ form the basis of the irreducible representation $T_{1u}$, while the components of the symmetric tensor $u$ transform as the symmetric square $[T_{1u}^2] = A_{1g} + E_g + T_{2g}$. Therefore, $u' = \nabla \otimes u$ transforms as the product $T_{1u} \times (A_{1g} + E_g + T_{2g}) = A_{2u} + E_u + 3T_{1u} + 2T_{2u}$. The components of the rank-three tensor $u'$ form the basis of a reducible representation. To find the irreducible combinations $u'_{\Lambda\lambda}(G)$, we apply the so-called Clebsch–Gordan decomposition of the group theory. Here, $\Lambda$ is the resultant irreducible representations $A_{2u}$, $E_u$, $3T_{1u}$, and $2T_{2u}$, $\lambda$ being their rows, and $G$, similar to the quantum number seniority in atomic spectra, labels the original term in the product, $T_{1u} \times A_{1g}$, $T_{1u} \times E_g$, or $T_{1u} \times T_{2g}$.

The left side of Equation (10) is a vector transforming as $T_{1u}$. Hence, of the seven representations $A_{2u}$, $E_u$, $3T_{1u}$, and $2T_{2u}$, we keep only the three vector representations $T_{1u}$, originating from the above products. We present components of the vectors $\mathbf{u}'(A_{1g})$, $\mathbf{u}'(E_g)$ and $\mathbf{u}'(T_{2g})$ by the expressions $u'_j(\Lambda) = \sum_{k,\lambda} (\partial u_{\Lambda\lambda}/\partial x_k) T_{1u}k\Lambda\lambda | T_{1u}j$, where $\langle \Lambda_1 \gamma_1 \Lambda_2 \gamma_2 | \Lambda\lambda \rangle$ are the Clebsch–Gordan coefficients. In these terms, Equation (10) takes the form similar to the one with susceptibility, $P_i = \sum_j \left[ f_{ij}(A_{1g}) u'_j(A_{1g}) + f_{ij}(E_g) u'_j(E_g) + f_{ij}(T_{2g}) u'_j(T_{2g}) \right]$,

$$P_i = \sum_\Lambda \sum_j f_{ij}(\Lambda) u'_j(\Lambda), \text{ with } f_{ij}(\Lambda) = \sum_{k,m,n,\lambda} f_{ikmn} \langle \Lambda\lambda | T_{1u}m T_{1u}n \rangle \langle T_{1u}j | T_{1u}k\Lambda\lambda \rangle \tag{11}$$

Instead of just one tensor $\varepsilon_0 \chi_{ij}$, it includes three tensors, $f_{ij}(A_{1g})$, $f_{ij}(E_g)$, and $f_{ij}(T_{2g})$.

The applied non-uniform strain induces a local electric field **E′** polarizing the crystal, so that the perturbation becomes $W = -p_0\left(E'_x \mathbf{C}_x + E'_y \mathbf{C}_y + E'_z \mathbf{C}_z\right)$ with $\mathbf{E}' = V_A \mathbf{u}'(A_{1g}) + V_E \mathbf{u}'(E_g) + V_T \mathbf{u}'(T_{2g})$. Here, $V_\Lambda$ (with $\Lambda = A_{1g}$, $E_g$, and $T_{2g}$) are the corresponding coupling constants. Assuming that the strain gradient is relatively weak, and involving the relations $\partial P_i/\partial u_j' = (\partial P_i/\partial E_j')(\partial E_j'/\partial u_j') = V_\Gamma(\partial P_i/\partial E_j')$, we get $f_{ij}(\Lambda) = f(\Lambda)\delta_{ij}$ with $\Lambda$ of the products $T_{1u} \times A_{1g}$, $T_{1u} \times E_g$ and $T_{1u} \times T_{2g}$. Quite similar to the procedure that leads to Equation (7), Equation (11) yields the following:

$$f(\Lambda) \approx \frac{p_0^2 V_\Lambda}{3a^3\Gamma} \frac{\sinh(3\Gamma\beta) + \sinh(\Gamma\beta)}{\cosh(3\Gamma\beta) + 3\cosh(\Gamma\beta)} \tag{12}$$

Thus, in cubic perovskites, among the 81 components of $f_{ijkl}$, only three reduced matrix elements $f(\Lambda)$ with $\Lambda = A_{1g}$, $E_g$, and $T_{2g}$ remain independent. In addition to the cubic symmetry-related

constraints, the flexotensor is invariant to index transpositions [59] $f_{ijkl} = f_{jikl}$ and $f_{ijkl} = f_{ilkj}$. Therefore, the three reduced matrix elements can be expressed in terms of just two components, $f(A_{1g}) = f_{1111}\sqrt{3}$, $f(E_g) = (f_{1111} - f_{1122})\sqrt{2}$, and $f(T_{2g}) = 2f_{1122}\sqrt{3}$ [60]. Accordingly, just two measurements are required to determine $f(\Lambda)$ experimentally. For example, to find the tetragonal component $f(E_g) = (2P_z\sqrt{2})(\partial u_{zz}/\partial z)^{-1}$, it is sufficient to measure the polarization that occurs under a non-uniform strain applied along the crystal axis [001], with the strain gradient $\partial u_{zz}/\partial z \neq 0$. Similarly, if the strain gradient is along the trigonal direction [111], one can find $f(T_{2g}) = P/u'(T_{2g})$.

The applied strain shifts the equilibrium positions of all ions in the unit-cell octahedron [$BO_6$]. Therefore, in Equation (12), the coupling constants $V_\Gamma$ are related to the vibronic coupling constant $F$ in Sections 2 and 3. For example, consider a tetragonal uniaxial strain along [001] with $u_q = u_{zz}$ and $u_e = (\sqrt{3}/2)(u_{xx} - u_{yy}) = 0$ or, equivalently, $u_{xx} = u_{yy} = -\tfrac{1}{3}u_q$ and $u_{zz} = \tfrac{2}{3}u_q$. It shifts the equilibrium position $\mathbf{R}^{(0)}_{\mathbf{n}j} = \langle X^{(0)}_{\mathbf{n}j}, Y^{(0)}_{\mathbf{n}j}, Z^{(0)}_{\mathbf{n}j}\rangle$ of the $j$th ion in the $\mathbf{n}$th unit cell to $\mathbf{R}'_{\mathbf{n}j} = \langle X^{(0)}_{\mathbf{n}j}(1 - \tfrac{1}{3}u_{\mathbf{n}\theta}), Y^{(0)}_{\mathbf{n}j}(1 - \tfrac{1}{3}u_{\mathbf{n}\theta}), Z^{(0)}_{\mathbf{n}j}(1 + \tfrac{2}{3}u_{\mathbf{n}\theta})\rangle$, where $j = 1, 2, \ldots, 7$ labels the seven ions in the octahedron [$BO_6$]. We need the site index $\mathbf{n}$ in the symmetry-adapted strain $u_{\mathbf{n}\theta}$ to include its non-uniform nature. In the perovskite crystal lattice, the adjacent octahedrons [$BO_6$] share a bridge oxygen atom. In the next-neighbor octahedron [$BO_6$]$_{\mathbf{n}+1}$ shifted in the direction [001] by one lattice constant $a$, due to the non-zero strain gradient, the symmetry-adapted coordinates are slightly different. Comparing the tetragonal deformation of the adjacent unit cells, let $u'_\theta = \Delta u_q/\Delta z = (u_{\mathbf{n}+1,q} - u_{\mathbf{n}q})/a$ where, as above, $a$ is the lattice constant. It follows that the non-uniform tetragonal deformation creates a dipolar distortion $Q_z \rightarrow Q_z + 0.1a^2 u'_\theta$ or, in other words, $\Delta Q_z = 0.1a^2 u'_\theta$. This gives $V_E = 0.1Fa^2 = 0.1Z_Bea^2 u'_\theta$. The corresponding energy increment $F\Delta Q_z = V_E u'_\theta = 0.1Fa^2 u'_\theta$ can be treated as being due to a local electric field $E'$ induced by the non-zero gradient $u'_\theta$. Then, $0.1Fa^2 u'_\theta = p_0 E'$ and, therefore, $E' = 0.1Fa^2 u'_\theta/p_0$. Correspondingly, the induced polarization is $P_z = \varepsilon_0\alpha E' = 0.1\varepsilon_0\alpha Fa^2 u'_\theta/p_0$.

The resultant component of the flexotensor, $f_{1111} = \partial P_z/\partial u'_\theta = 0.1\varepsilon_0\alpha Fa^2/p_0$, is proportional to the dielectric susceptibility $\alpha$. For BaTiO$_3$, with the experimental value of $\alpha$ of the order of $2 \times 10^4$, we find $f_{1111} = 0.1\varepsilon_0\alpha Fa^2/p_0 \approx 0.66$ μC/m, which is around 2600 times greater than was found in [58] and much nearer to the experimental value. Besides, with the orientational polarizability, as it follows from the outlined here PJTE theory, the flexoelectric factor is positive (in accordance with experimental data), whereas without the latter, it emerges as negative [58]. For other limiting conditions, rough estimates [51–53] yield even higher orientational contributions to the flexoelectric coefficients.

## 5. Multiferroicity in ABO$_3$ Perovskites with B($d^n$) Configurations

The vibronic theory of ferroelectricity in ABO$_3$ perovskite crystals based on PJTE-induced local dipolar distortions in the B centers of the perovskite crystal was extended to formulate the necessary condition of coexisting magnetic and ferroelectric (multiferroicity) properties [3,4]. "Multiferroicity implies that the ferroelectric crystal, which is a dielectric, has also a nonzero magnetic moment, meaning unpaired electrons. In the ferroelectric BaTiO$_3$ the $d^0$ configuration of the Ti$^{4+}$ ion has no unpaired electrons, and attempts to obtain ferroelectricity in perovskites with $d^n$, $n>0$, configurations of the transition metals B ions were unsuccessful for a long time. This prompted some authors to term the situation as a ferroelectric "$d^0$ *mystery*" [61–63], accompanied by a conclusion that nonzero spin states are detriment to ferroelectric polarization. However, more recently quite a number of ferroelectrics-multiferroics, mostly perovskites with configurations $d^3$–$d^7$, were obtained and studied" [3,4].

The origin of these special properties of perovskite ferroelectrics with $d^n$ configurations does not follow directly from displacive theories and had no general explanation for a long time. The vibronic (PJTE) theory of ferroelectricity outlined above elucidates the role of spin states in the local polar instability, explains the origin of perovskite multiferroics with proper ferroelectricity, and formulates the necessary conditions that ABO$_3$ perovskites with a magnetic $d^n$ configuration of the B ion may be ferroelectric [2–4]. Moreover, because of the spin implication, the multiferroics conditions that

emerge from the PJTE for $d^n$ ions with $n$ = 3, 4, 5, 6, and 7 are directly influenced by the well-known transition metal high-spin/low-spin crossover, resulting in the coexistence of three phenomena: ferroelectricity (FE), magnetism (M), and spin crossover (SCO). This, in turn, leads to a quite novel phenomenon, magnetic-ferroelectric (multiferroics) crossover (MFCO), creating a rich variety of possible magnetoelectric and related effects [3,4]. The vibronic (PJTE) theory, exclusively, reveals the role of spin in the spontaneous polarization of crystals.

Referring to the typical MO energy scheme for the octahedral cluster $BO_6^{8-}$ discussed above (Section 2) and shown in Figure 2 for the MO electron population of the $d^0$ configuration, e.g., when B ≡ Ti, we see that the HOMO in this case is $t_{1u}$, which is a three-fold degenerate odd-parity linear combination of mostly oxygen $p_\pi$ orbitals, while the LUMO is $t_{2g}$, mostly atomic three $d_\pi$ orbitals of the transition metal ion B, and the next excited MO is the double degenerate one $e_g$ (the non-bonding oxygen $b_{1g}$ MO is not shown as irrelevant). Using the arrows up and down to indicate the two spin states, we find for the $d^0$ case the HOMO configuration $(t_{1u})^6 = (t_{1u}\downarrow)^3(t_{1u}\uparrow)^3$, with the energy term $^1A_{1g}$. The excited state with opposite parity is formed by the one-electron, $(t_{1u}\uparrow) \rightarrow (t_{2g}\uparrow)$ or $(t_{1u}\uparrow) \rightarrow (e_g\uparrow)$, excitation, resulting in the lowest excited odd-parity term $^1T_{1u}$ at the energy gap $2\Delta$. In this case, the PJTE at the B center, under the condition of instability $\Delta < 8F^2/K_0$ (Equation (3)), produces a polar displacement of the B atom along [111]-type directions, which triggers the ferroelectric polarization (Section 3).

For other $d^n$ configurations of the B ion instead of the $d^0$ one, the electronic structure of the ground and low-lying states changes drastically, and so does the possibility of formation of close in energy ground and excited states of opposite parity but equal multiplicity. In particular, for the B($d^1$) ions, the HOMO becomes $(t_{1u}\downarrow)^3(t_{1u}\uparrow)^3(t_{2g}\uparrow)^1$ with the term $^2T_{2g}$, while the LUMO (taking into account Hund's rule) is $(t_{1u}\downarrow)^2(t_{1u}\uparrow)^3(t_{2g}\uparrow)^2$ with the lowest excited odd-parity term $^4T_{1u}$. Hence, the two closest terms of different parity possess different spin multiplicity, and hence they do not mix by the vibronic coupling; the latter does not contain spin operators [9,10]. In principle, higher in energy electronic configurations of opposite parity with the same spin as the ground state one are quite possible, but they are at much larger energy gaps $\Delta$ and therefore less appropriate to satisfy the condition of instability (3) (numerical estimates show that the condition (3) may be very restricting). For the next possible $d^2$ configurations of the B ion, the two lowest terms of opposite parity are $^2T_{2g}$ and $^5T_{1u}$, which, again, do not satisfy the condition for the PJTE dipolar instability.

Moving to the case of B($d^3$) configuration, we find that the HOMO becomes $(t_{1u}\downarrow)^3(t_{1u}\uparrow)^3(t_{2g}\uparrow)^3$ with the ground state term $^4A_{1g}$, and in the low-spin (LS) conditions of the strong ligand fields (sufficiently large $t_{2g}$-$e_g$ separation in Figure 2), the LUMO is $(t_{1u}\downarrow)^2(t_{1u}\uparrow)^3(t_{2g}\uparrow)^3(t_{2g}\downarrow)^1$ with the lowest odd-parity term $^4T_{1u}$. It follows that in perovskites with B($d^3$) ions in LS conditions of sufficiently strong ligand fields, the situation becomes again favorable for the PJTE and polar distortions. However, in this case, distinguished from the $d^0$ case, the system possesses also a magnetic moment created by three unpaired electrons. However, if the ligand field is weak and the separation $t_{2g}$-$e_g$ is small, the high-spin (HS) arrangement of the excited electronic configuration takes place, and the excitation electron occupies the $e_g\uparrow$ orbital instead of $t_{2g}\downarrow$; the LUMO configuration under Hund's rule becomes $(t_{1u}\downarrow)^2(t_{1u}\uparrow)^3(t_{2g}\uparrow)^3(e_g\uparrow)^1$ with the lowest odd-parity state $^6T_{1u}$. Here, again, there is no PJTE on dipolar distortions and no ferroelectric instability. This is one of the examples which shows explicitly how the spin states interfere directly in the possible local polar displacement and ferroelectricity.

Repeating the above procedure for all the other $d^n$ configurations with $n$ = 0, 1, 2, ..., 10, it was shown that in perovskite $ABO_3$ crystals, only B ions with configurations $d^0$, $d^3$-LS, $d^4$-LS, $d^5$-LS, and HS, $d^6$-HS and intermediate-spin (IS), $d^7$-HS, $d^8$, and $d^9$ can, in principle, produce multiferroics, provided that the criterion of instability (3) is satisfied (see Table 2). If the contribution of higher excited states can be ignored, transition metal ions B with configurations $d^1$, $d^2$, $d^3$-HS $d^4$-HS, $d^6$-LS, $d^7$-LS, and $d^{10}$ are not expected to produce multiferroics under this mechanism of proper ferroelectricity. Experimentally observed perovskite multiferroics with such B ions, for example, $Mn^{4+}(d^3)$, $Cr^{3+}(d^3)$,

$Mn^{3+}(d^4)$, $Fe^{3+}(d^5)$, $Fe^{2+}(d^6)$, $Co^{2+}(d^7)$, etc., fit well with this conclusion; there are no multiferroics with $d^0$, $d^1$, $d^2$, and $d^{10}$ configurations.

**Table 2.** Necessary conditions that $ABO_3$ perovskites with the electronic $d^n$ configuration of the B ion possess both ferroelectric and magnetic properties simultaneously; EC = electronic configuration, GS = ground state, LUES = lowest odd-parity excited state, FE = ferroelectric, MM = magnetic, MF = multiferroic, LS = low-spin, HS = high-spin; IS = intermediate spin; $(t_{1u})^6 = (t_{1u}\downarrow)^3 (t_{1u}\uparrow)^3$; $(t_{1u})^5 = (t_{1u}\downarrow)^2 (t_{1u}\uparrow)^3$; $(t_{2g})^6 = (t_{2g}\uparrow)^3 (t_{2g}\uparrow)^3$; $(e_g)^4 = (e_g\uparrow)^2 (e_g\downarrow)^2$ [3].

| $d^n$ | Example | HOMO EC and GS Term | LUMO EC and LUES Term | FE | MM | MF |
|---|---|---|---|---|---|---|
| $d^0$ | $Ti^{4+}$ | $(t_{1u})^6$, $^1A_{1g}$ | $(t_{1u})^5 (t_{2g}\uparrow)^1$, $^1T_{1u}$ | yes | no | no |
| $d^1$ | $Ti^{3+}$, $V^{4+}$ | $(t_{1u})^6 (t_{2g}\uparrow)^1$, $^2T_{2g}$ | $(t_{1u})^5 (t_{2g}\uparrow)^2$, $^4T_{1u}$ | no | yes | no |
| $d^2$ | $V^{3+}$, $Cr^{4+}$ | $(t_{1u})^6 (t_{2g}\uparrow)^2$, $^3T_{2g}$ | $(t_{1u})^5 (t_{2g}\uparrow)^3$, $^5T_{1u}$ | no | yes | no |
| $d^3$, LS | $Cr^{3+}$, $Mn^{4+}$ | $(t_{1u})^6 (t_{2g}\uparrow)^3$, $^4A_{2g}$ | $(t_{1u})^5 (t_{2g}\uparrow)^3 (t_{2g}\downarrow)^1$, $^4T_{1u}$ | yes | yes | yes |
| $d^3$, HS | | $(t_{1u})^6 (t_{2g}\uparrow)^3$, $^4A_{2g}$ | $(t_{1u})^5 (t_{2g}\uparrow)^3 (e_g\uparrow)^1$, $^6T_{1u}$ | no | yes | no |
| $d^4$, LS | $Mn^{3+}$, $Fe^{4+}$ | $(t_{1u})^6 (t_{2g}\uparrow)^3 (t_{2g}\downarrow)^1$, $^3T_{2g}$ | $(t_{1u})^5 (t_{2g}\uparrow)^3 (t_{2g}\downarrow)^2$, $^3T_{1u}$ | yes | yes | yes |
| $d^4$, HS | | $(t_{1u})^6 (t_{2g}\uparrow)^3 (e_g\uparrow)^1$, $^5T_{2g}$ | $(t_{1u})^5 (t_{2g}\uparrow)^3 (e_g\uparrow)^2$, $^7T_{1u}$ | no | yes | no |
| $d^5$, LS | $Mn^{2+}$, $Fe^{3+}$ | $(t_{1u})^6 (t_{2g}\uparrow)^3 (t_{2g}\downarrow)^2$, $^2T_{2g}$ | $(t_{1u})^5 (t_{2g})^6$, $^2T_{1u}$ | yes | yes | yes |
| $d^5$, HS | | $(t_{1u})^6 (t_{2g}\uparrow)^3 (e_g\uparrow)^2$, $^6A_{1g}$ | $(t_{1u})^5 (t_{2g})^4 (e_g\uparrow)^2$, $^6T_{1u}$ | yes | yes | yes |
| $d^6$, LS | $Fe^{2+}$, $Co^{3+}$ | $(t_{1u})^6 (t_{2g})^6$, $^1A_{1g}$ | $(t_{1u})^5 (t_{2g})6 (e_g\uparrow)^1$, $^3T_{1u}$ | no | no | no |
| $d^6$, $IS_1$ | | $(t_{1u})^6 (t_{2g})^5 (e_g\uparrow)^1$, $^3T_{1g}$ | $(t_{1u})^5 (t_{2g})6 (e_g\uparrow)^1$, $^3T_{1u}$ | yes | yes | yes |
| $d^6$, $IS_2$ | | $(t_{1u})^6 (t_{2g})^5 (e_g\uparrow)^1$, $^3T_{1g}$ | $(t_{1u})^5 (t_{2g})^5 (e_g\uparrow)^2$, $^5T_{1u}$ | no | yes | no |
| $d^6$, HS | | $(t_{1u})^6 (t_{2g})^4 (e_g\uparrow)^2$, $^5T_{2g}$ | $(t_{1u})^5 (t_{2g})^5 (e_g\uparrow)^2$, $^5T_{1u}$ | yes | yes | yes |
| $d^7$, LS | $Co^{2+}$, $Ni^{3+}$ | $(t_{1u})^6 (t_{2g})^6 (e_g\uparrow)^1$, $^2E_g$ | $(t_{1u})^5 (t_{2g})^6 (e_g\uparrow)^2$, $^4T_{1u}$ | no | yes | no |
| $d^7$, HS | | $(t_{1u})^6 (t_{2g})^5 (e_g\uparrow)^2$, $^4T_{2g}$ | $(t_{1u})^5 (t_{2g})^6 (e_g\uparrow)^2$, $^4T_{1u}$ | yes | yes | yes |
| $d^8$ | $Ni^{2+}$, $Cu^{3+}$ | $(t_{1u})^6 (t_{2g})^6 (e_g\uparrow)^2$, $^3A_{1g}$ | $(t_{1u})^5 (t_{2g})^6 (e_g)^3$, $^3T_{1u}$ | yes | yes | yes |
| $d^9$ | $Cu^{2+}$ | $(t_{1u})^6 (t_{2g})^6 (e_g)^3$, $^2E_g$ | $(t_{1u})^5 (t_{2g})^6 (e_g)^4$, $^2T_{1u}$ | yes | yes | yes |
| $d^{10}$ | $Zn^{2+}$ | $(t_{1u})^6 (t_{2g})^6 (e_g)^4$, $^1A_g$ | $(t_{1u})^5 (t_{2g})^6 (e_g)^4 (ns\uparrow)^1$, $^3T_{1u}$ | no | no | no |

Consider now that some $d^n$ ions with $n$ = 3, 4, 5, 6, and 7, dependent on the ligands of the octahedral environment, may produce two types of magnetic centers, high-spin (HS) and low-spin (LS), and in the $d^6$ case, there maybe also intermediate spin (IS) states ($d^3$ has two spin configurations in the one-electron excitation). According to the analysis [2–4], only $d^5$ systems follow the necessary condition of potential multiferroics in both spin states, but the PJTE conditions of instability and the magnetic moments are different in these two cases. For $d^3$, $d^4$, $d^6$, and $d^7$ ions, only one of the two spin states may serve as a candidate of potential multiferroics.

On the other hand, in many cases, the two spin states are close in energy, producing the well-known phenomenon of transition metal spin crossover (SCO), in which case the system can be relatively easily transferred from one spin state to another by external perturbations like heat, light, and magnetic fields [3,4]. As shown above, the change in the spin state changes also the possibility of ferroelectric polarization; hence, the SCO in some perovskite crystals is simultaneously a magnetic–ferroelectric (multiferroic) crossover (MFCO). This coexisting magnetic, ferroelectric, and spin-crossover phenomenon opens a variety of new possibilities to manipulate the properties of the system with novel functionalities to electronics and spintronics [3,4]: (1) For $d^3$ and $d^4$ ($Cr^{3+}$, $Mn^{4+}$, $Mn^{3+}$, $Fe^{4+}$, etc.) ferroelectrics in the LS state in conditions of MFCO, magnetic fields facilitate the LS→HS transition that destroys the ferroelectricity (and multiferroicity), while an electric field in the HS non-ferroelectric state may transfer the system to the ferroelectric (multiferroic) LS state, thus realizing electric demagnetization; (2) For $d^5$ ferroelectrics in conditions of MFCO, if the

ferroelectricity is (most probably) different in the two spin states, an electric field may change the spin state (electric magnetization or demagnetization); (3) For $d^6$ and $d^7$ ($Fe^{2+}$, $Co^{3+}$, $Co^{2+}$, $Ni^{3+}$, etc.) in the non-ferroelectric LS state under conditions of MFCO, magnetic fields facilitate the LS→HS transition that induces ferroelectricity and hence multiferroicity in a strong magnetoelectric effect (the $d^6$ LS state is nonmagnetic); in the non-ferroelectric LS state in MFCO conditions, an electric field may transfer the system to the multiferroic state (electric magnetization); (4) The SCO phenomenon is well known to be influenced also by stress, heat, light, and cooperative effects in crystals [46], hence these perturbations can be used to manipulate the MFCO and all the consequent properties including those mentioned above. The dependence of the MFCO on pressure adds a ferroelastic order to the magnetic and ferroelectric ones; (5) There is already a long history of attempts to use transition metal SCO systems as units of magnetic bistability. The difficulty is in the fast relaxation (short lifetime) of the higher in energy spin state [64–66]. By choosing a system in the MFCO condition, one can increase the lifetime of the excited dipolar (multiferroic) state by applying an external electric field; (6) An important feature of the revealed MFCO is that it is of local origin and hence it does not necessarily require strong cooperative interactions, meaning that, in principle, it may take place as a magnetic-dipolar effect in separate molecular systems, clusters, thin films, etc., provided that the condition of instability (3) is obeyed.

In addition to the mentioned above examples that confirm the (outlined by the PJTE theory) necessary conditions of multiferroicity, several papers [67–69] reported observing also the predicted [3,4] magnetic–ferroelectric spin-crossover effect, while in [69], it is shown that in $BiCoO_3$, the ferroelectric polarization is greatly enhanced when the $Co^{3+}$ ion is in the high-spin state, as compared to the nonmagnetic state with the $Co^{3+}$ ion in the low-spin configuration. They demonstrated also the predicted electric magnetization [3] (see point 3 above), when, by means of induced polarization, the spin state changes from low-spin (S = 0) to high-spin (S = 2). The authors concluded that, contrary to the widespread belief, "*unpaired electron spins actually drive ferroelectricity, rather than inhibit it, which represents a shift in the understanding of how ferroelectricity and magnetism interact in perovskite oxides*" [69].

## 6. Origin of Polar Nanoregions and Relaxor Properties

The PJTE theory recently solved also another problem of ferroelectric crystals, the origin of polar nanoregions (PNR) producing relaxor properties [7], which remained unsolved (inconclusive) in spite of over 60 year of studies (see [70–74] and references therein). PNR in the non-polarized, paraelectric phases are observed arguably in all perovskite ferroelectrics. They are formed spontaneously in the paraelectric phases of ferroelectrics above the Curie temperature $T_C$ (where the bulk crystal is cubic and no polarization is expected) in the form of small islands (nanoregions), containing a limited number $n$ of unit cells, their size decreasing with increasing temperature $T_n$-$T_C$. Above the so-called Burns temperature $T_B$, the PNR disappear, and the crystal becomes regular, paraelectric. PNR also disappear under sufficiently strong electric fields. Significantly above $T_C$, the paraelectric phase remains ergodic. With cooling, PNR grow in size, and at a temperature $T_f$ (closer to $T_C$, $T_C < T_f < T_B$), relaxor properties change again to a non-ergodic, glass-like state, which then undergoes the phase transition to the tetragonal polarized state at $T = T_C$. Distinguished from dipole glasses, this non-ergodic state of the crystal can be irreversibly transformed into a regular polarized state by strong external electric fields. These relaxor properties of ferroelectrics influence all their main properties, with direct applications in materials science.

Attempts to explain the formation of PNR as being due to basic structural disorder, or "random fields" produced by the differences in the active centers (especially in mixed perovskites), as well as by other crystal imperfections (see [70–74]), failed because they do not explain the origin of temperature-dependent size effects—in particular, the disappearance of PNR above $T_B$, or under polarizing electric fields. Moreover, PNR are present in ferroelectric perovskites, but they do not show up in very similar non-ferroelectric crystals with structural phase transitions.

The vibronic (PJTE) theory of ferroelectricity explains also the origin of PNR and relaxor properties of ferroelectric crystals [7] as due to the described above local dynamics of the dipolar distortions at the B centers, which are fully disordered (uncorrelated) in the paraelectric phase and partially ordered (correlated) in the ferroelectric phases. At temperatures above $T_C$, the Helmholtz free energy of the cubic phase $\Phi_{cub}$ is lower than its value in the tetragonal phase $\Phi_{tetr}$. Accordingly, as $\Phi = U - TS$, where U is the internal energy and S is the entropy, in the temperature interval $T_C < T < T_B$, we have

$$T(S_{cub} - S_{tetr}) > U_{cub} - U_{tetr} \tag{13}$$

This means that at $T > T_C$, the energy gain $\Delta U = U_{cub} - U_{tetr}$ in lowering the potential energy from (average) cubic (in the paraelectric phase) to (average) tetragonal (in the PNR) is not enough to compensate the corresponding entropy loss, $T(S_{cub} - S_{tetr})$. Therefore, no displacive theory can explain their formation and specific properties.

Based on the basic features of the vibronic PJTE theory of ferroelectricity, outlined above, the local polarized isles (polar nanoregions, PNR, Figure 9) with a limited, relatively small number $n$ of centers are formed in a non-equilibrium self-assembly process of alignment of the local dipolar displacements [7]. As shown below, at $T_n > T_C$, the inequality (13) is compensated by the transformation of a part of the tetragonal potential energy $\Delta U_{tetr}$ of the PNR into the work of the formation of its surface $W_n$ and transfers heat Q to the environment. The gain of energy in the formation of the $n$-center PNR $\Delta U_n = U_n^{(cub)} - U_n^{(tetr)}$ consists of two contributions: $\Delta U_n^{(in)}$, the energy of the internal centers, i.e., the energy of ordering the $n$ dipoles inside the PNR (all oriented, on average, along the polarization direction), and $\Delta U_n^{(surf)}$, the energy of the centers in the surface layer (the "domain wall"), which are influenced by the neighbor disordered centers of the cubic phase (Figure 9). Denoting by $n$ and $n'$ the total number of centers in the PNR and in its border surface, respectively, we get the following estimates for the ordering energies based on the mentioned above mean-field approximation (Section 3): $\Delta U_n^{(in)} = (n - n') \Delta U_0$ and $\Delta U_n^{(surf)} = n'(\Delta U_0 - g)$. Here, $\Delta U_0$ is the per-unit energy gain by tetragonal ordering and $g$ is the loss in this energy by the surface centers because of them being in a lower mean-field, $E' < E$, of the environment; the mean-field contribution from a part of their environment, by the disordered centers of the bulk cubic phase, is zero. A rough estimate yields $g \gtrsim (1/6) \Delta U_0$ [7].

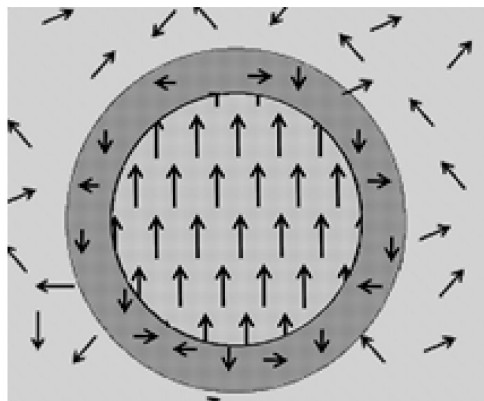

**Figure 9.** Polarized nanoregion inside the cubic perovskite with spherical form (center) and a border layer (dark) at temperatures $T$ above the Curie one TC, but below the Burns temperature $T_B$, $T_C < T < T_B$. Arrows indicate the direction of the local averaged dipole moments, which are tetragonally ordered inside the PNR and fully disordered in the cubic phase. For the role of the PNR surface layer, see the text (reproduced with permission from [7]. Copyright 2018, American Physical Society).

For the system in a thermostat under consideration, the exchange of energy with the environment may take place without free energy conservation, $\Phi_i \neq \Phi_f$, the total energy balance being preserved by compensation of heat transfer, $Q = T(S_f - S_i) = T\Delta S$, with internal energy change $\Delta U$ and

mechanical work of internal forces, $W_{\text{intern}}$. Similar to other processes with heat (entropy) transfer, the formation of PNR is a non-equilibrium thermodynamic process, described by Gibbs free energy change, $\Delta G = \Delta H - T\Delta S$, where $\Delta H$ is the change in enthalpy H. With growing size of PNR, their Gibbs free energy decreases and reaches a minimum value when $n$ satisfies the condition of thermodynamic equilibrium with the environment. At this point, the shape-restoring force, $-dG/dr$, is close to zero, while according to the fist law of thermodynamics, $\Delta U = Q - W_{\text{intern}}$. This determines the size of the PNR with the number of units $n$ at a given temperature $T_n$. Some relatively simple estimates based on the PJTE-induced APES in perovskite crystals briefly outlined above (Sections 2 and 3) yield the following temperature dependence of the size of PNR [7]:

$$\frac{T_n}{T_{\text{C}}} = 1 + \frac{4.8g}{\Delta U_0} n^{-1/3}, \quad T_n = T_{\text{C}}\left(1 + \frac{4.8g}{\Delta U_0} n^{-1/3}\right) \tag{14}$$

or by introducing the crystal constant $A = 36\pi\left(\frac{g}{\Delta U_0}\right)^3$, we obtain (Figure 10) the following:

$$n = A\left(\frac{T_n}{T_{\text{C}}} - 1\right)^{-3} \tag{15}$$

Figure 10 shows also the size of PNR by its diameter $D$, which is obtained under the assumption of its (most probably) approximately spherical form.

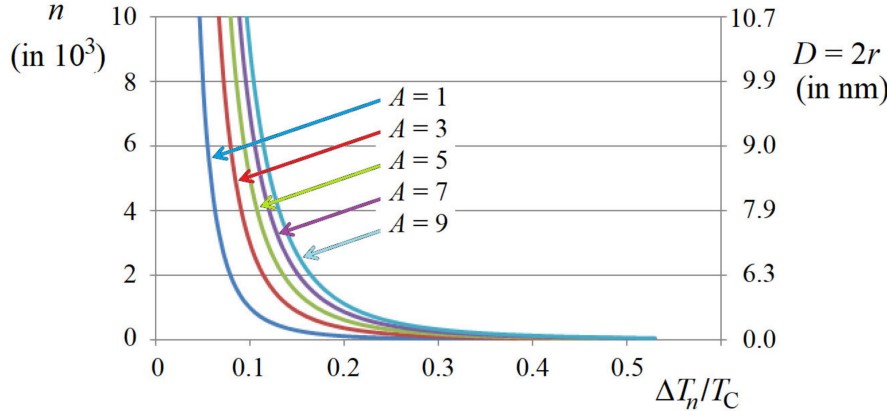

**Figure 10.** (Color online). Temperature dependence of the PNR size shown by the number of centers $n$ (in $10^3$, left scale), and the diameter of its spherical form $D$ (in nm, right scale), as a function of the crystal parameter $A$ (reproduced with permission from [7]. Copyright 2018, American Physical Society).

The most important cubic $(T_n - T_{\text{C}})^{-3}$ dependence of the PNR size is confirmed experimentally. Subject to the power "$-3$", the increase in the size of the PNR with decrease in temperature is very fast, especially at smaller increments $\Delta T_n/T_{\text{C}} \approx 0.1$–$0.2$. This explains the formation of the nonergodic "glassy" state of relaxor ferroelectrics [7]. The estimates leading to Equation (14) show also [7] that, albeit small, an initial trigger (fluctuation) is needed to start the growth of a PNR. It clarifies the role of crystal imperfections or structural irregularities (particularly in mixed perovskites) in enhancing the formation of PNR. Another important point is that the formation of the PNR in time is strongly dependent on the speed of cooling [7].

With the PNR included, we can revisit the polarizability of ferroelectric perovskites. Under an external electric field **E**, the PNR turns as a whole, like a dipolar molecule, aligning with **E,** its total

dipole moment $\langle p \rangle \approx np_0/\sqrt{6}$. Applying the Debye–Langevin equation to the ensemble of these "molecules," we find the actual PNR-induced polarizability,

$$\alpha = \frac{Nn^2 p_0^2}{18a^3 \varepsilon_0 kT} = \frac{NA^2 p_0^2}{18a^3 \varepsilon_0 kT}\left(\frac{T}{T_C} - 1\right)^{-6} \tag{16}$$

Here, $N$ is the concentration of PNR, close to the number of precursors (imperfections) in the crystal lattice that trigger their formation. We neglect the contribution of the applied external field to $N$. Notably, as follows from Equation (16), the PNR contribution to the polarizability of perovskite relaxor ferroelectrics manifests non-Curie–Weiss behavior: due to the strong temperature dependence of the size of PNR, ferroelectric perovskites have a much stronger temperature dependence of $\alpha$, proportional to $(T - T_C)^{-6}$, which may serve as an indication of the PNR contribution (experimental measured polarizabilities contain both PNR and bulk crystal contributions). Note, however, that the employed above Debye–Langevin equation is not applicable for temperatures near the phase transition where the crystal with PNR becomes non-ergodic and glassy-like, and the formation of the latter depends also on the speed of cooling [7]. For $BaTiO_3$, at temperatures above the glassy state, assuming (as above) that $p_0 \approx 7$ D, $a \approx 4$ Å and $T \approx 450$ K, with $N$ of the order of 0.1% and $A \approx 5$ [7], we come to a rather realistic estimate, $\alpha(PNR) \approx 5.6 \times 10^4$ [71–74].

## 7. Conclusions

The main conclusion that emerges from the outlined above studies of perovskite crystals, notably $BaTiO_3$, is that its spontaneous polarization and all relevant properties are triggered by the local PJTE. This conclusion is in drastic contrast to the long dominated "displacive" theories, which are based on the assumption that the spontaneous polarization of the crystal occurs due to the compensation of the local repulsions (in the dipolar distortions) with the long-range dipole–dipole attractions. The predicted by the PJTE local dipolar distortions in $BaTiO_3$ are multiply confirmed experimentally; they take place in all its phases irrelevant to the phase transitions. The latter occur as a result of the temperature-dependent ordering of the local dipolar displacements. Remarkably, this basic finding in perovskite ferroelectrics affects all their main properties and predicts novel features in interaction with external perturbation, important in applications. The latter includes a qualitatively novel property, the orientational polarization of solids, which (similar to the orientational polarization of polar liquids) is orders of magnitude larger than their displacive polarization. The possible presence of ready-made dipoles in solids which, above "freezing temperature", behave similarly to those in polar liquids was first suggested by P. Debye more than a century ago but revealed only by the vibronic theory. Extension of the PJTE theory to other perovskites and other crystal structures seems to be an up-to-date problem.

Another conclusion from these works is related to the general approaches to the study of solid-state problems. The overwhelming way to explore crystal structures and their properties is to start with revealing the cooperative electronic and nuclear motions characterized by electron and phonon bands formed by equivalent atoms or atomic groups, which are uniformly distributed with translation symmetry. The vibronic (PJTE) theory of ferroelectricity shows (convincingly) that this approach to the problem may be inadequate, losing the main properties of the system. The traditional electron and phonon band approach to solid-state problems fails when the translation symmetry groups are equivalent in average, but not equivalent at each moment in time. This takes place when there are additional local nuclear dynamics induced by the JTE or PJTE, which are uncorrelated (before their cooperative ordering). There are no general theories or computer programs to reveal the band structure of such systems. Our works are the first to reveal this problem and to handle it in the particular case of local PJTE that triggers ferroelectricity and related properties in perovskite crystals.

A short note about our meetings with K. A. Muller and his visits to USSR is available on Supplementary Materials

**Supplementary Materials:** A short note about our meetings with K. A. Muller and his visits to USSR is available online at http://www.mdpi.com/2410-3896/5/4/68/s1.

**Author Contributions:** Conceptualization, I.B.B. and V.P.; Data curation, V.P.; Investigation, I.B.B. and V.P.; Methodology, I.B.B. and V.P.; Project administration, I.B.B. All authors have read and agreed to the published version of the manuscript.

**Funding:** This research received no funding.

**Conflicts of Interest:** The authors declare no conflict of interest.

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
