# Peer review of "Perovskite Crystals: Unique Pseudo-Jahn–Teller Origin of Ferroelectricity, Multiferroicity, Permittivity, Flexoelectricity, and Polar Nanoregions"

_condensedmatter, doi:10.3390/condmat5040068_

Round 1

Reviewer 1 Report

This manuscript presents an excellent summary of the outstanding contributions that the authors have made in the theory of ferroelectricity and multiferroic properties of perovskites since the mid sixties. The extension of the Jahn-Teller effect concept to the case of electronic non-degeneracy (pseudo Jahn-Teller effect) is show to be the basic step in the theory, that replaces the traditional displacive mechanism of ferroelectricity with that of orientational polarization in solids. The work refers to the dipolar distortions of the octahedral units BO6 in ABO3 perovskites and their coupling, eventually leading to ferroelectric phases. The rotational degree of freedom of BO6 polarization realizes a local pseudo-Jahn-Teller effect, similar to the Jahn-Teller effect but in the absence of electronic degeneracy in the undistorted configuration The authors show that the mutual interaction among rotating dipoles may induce in the paraelectric phase polar nanoregions and relaxor properties, and that the interaction with external field may induce an orientational polarization of the solid. Formally the pseudo-JT transition is for rotational symmetry what the Peierls transition is for translational symmetry, but with a quasi-degeneracy in the undistorted phase.

The authors recognize that there is no conflict with Bussmann et al theory of the instability driven by a soft mode. A natural question  that the authors could answer concerns the order parameter depicted in Fig. 7: what is the critical exponent predicted by the pseudo-JT model and how does it compare to the experimental ones?

Apart this minor point, this is an important review, worth being published in this special issue dedicated to Karl-Alex Müller.

I would only request that the authors, besides commenting on the above question, find a way to have the English of the manuscript improved. In particular English orthography and syntax need to be fixed, e.g., the abundant misuse of semicolons, or sentences like “the predicted by the vibronic theory order-disorder nature of the phase transitions in ferroelectric perovskites” for “the order-disorder nature of the phase transitions in ferroelectric perovskites predicted by the vibronic theory”. Another example: “the induced by the PJTE eight equivalent off-center positions of the atom B” for “the PJTE-induced eight equivalent off-center positions of the atom B…” or “the eight equivalent off-center positions induced by the PJTE of the atom B”, etc.

Author Response

We considered all the reviewer's comments.

The data in Fig. 7 were obtained by numerical calculations with a computer program, so no analytical constants are available.

Reviewer 2 Report

The article summarizes the results of many years of research on the phenomenon of spontaneous polarization of perovskite crystals. The authors provide numerous arguments in favor of a theory that considers pseudo-Jahn-Teller effect in the transition metals as a source of spontaneous polarization, which is the cause of the occurrence of ferroelectricity and related phenomena: multiferroicity, flexoelectricity and polar nanoregions. The local dipole distortions described by this theory in BaTiO3 are confirmed experimentally for all of its phases. The theory clarifies also the role of spin in the spontaneous polarization. The author also describes a new interesting property - orientational polarization in solids. The authors argue that this phenomenon has not yet been discovered, although P. Debye mentioned it a long time ago. A detailed and qualified review of the literature on this topic is also of great value to specialists.

Author Response

Thank you, no comments.

This manuscript is a resubmission of an earlier submission. The following is a list of the peer review reports and author responses from that submission.